# SIGNAL IN THE NOISE: POLYSEMANTIC INTERFERENCE TRANSFERS AND PREDICTS CROSS-MODEL INFLUENCE

**Bofan Gong[1],\*  Shiyang Lai[2],\*  Dawn Song[3],  James Evans[2]**
[1]Independent Scholar, [2]University of Chicago, [3]UC Berkeley
`bfangong@gmail.com, shiyanglai@uchicago.edu`

## ABSTRACT

*Polysemanticity* is pervasive in language models and remains a major challenge for interpretation and model behavioral control. Leveraging sparse autoencoders (SAEs), we map the polysemantic topology of two small models (`Pythia-70M` and `GPT-2-Small`) to identify SAE feature pairs that are semantically *unrelated* yet exhibit interference within models. We intervene at four foci (prompt, token, feature, neuron) and measure induced shifts in the next-token prediction distribution, uncovering polysemantic structures that expose a systematic vulnerability in these models. Critically, interventions distilled from *counterintuitive* interference patterns shared by two small models transfer reliably to larger instruction-tuned models (`Llama-3.1-8B/70B-Instruct` and `Gemma-2-9B-Instruct`), yielding predictable behavioral shifts without access to model internals. These findings challenge the view that polysemanticity is purely stochastic, demonstrating instead that interference structures generalize across scale and family. Such generalization suggests a convergent, higher-order organization of internal representations, which is only weakly aligned with intuition and structured by latent regularities, offering new possibilities for both black-box control and theoretical insight into human and artificial cognition. Code and data are available here.

## 1 INTRODUCTION

*Polysemanticity* refers to the phenomenon in which individual neurons or groups of neurons in neural networks often encode a greater number of distinct features or concepts than the number of neurons involved. This property becomes increasingly prevalent as models scale and has been shown to enhance learning performance (Wang et al., 2024; Marshall & Kirchner, 2024; Oikarinen & Weng, 2024b). Anthropic's work on *superposition* builds on prior insights, showing that large transformer models encode more features than neurons by using linear combinations of activations. This mechanism sacrifices monosemanticity but significantly improves model capability (Elhage et al., 2022). Mathematical analyses reveal that polysemantic neurons enable networks to represent exponentially more features compared to monosemantic approaches (Elhage et al., 2022).

Nevertheless, this representational efficiency comes with trade-offs. Most significantly, it complicates model interpretability, as entangled representations obscure how human-understandable concepts are encoded within the model's internal structure. One mechanistic approach to address this challenge is the use of sparse autoencoders (SAEs), which aim to disentangle superimposed features by learning sparse, higher-dimensional representations of model activations. SAEs enable the extraction of interpretable, monosemantic features, where each SAE neuron ideally corresponds to a single concept (Bricken et al., 2023; Templeton et al., 2024)[1]. Recent work has shown that SAE-derived features exhibit a degree of universality across different LLMs (Lan et al., 2024), suggesting the existence of fundamental patterns in how neural networks encode meaning. This consistency hints at the emergence of shared semantic topologies that persist across architectures and training regimes, raising profound questions about whether these patterns are merely computational artifacts or reflections of latent semantic regularities (Huh et al., 2024). Except for SAEs, a broader range of interpretability techniques is emerging simultaneously (Chang et al., 2025; Dunefsky et al., 2024).

---

\*Equal contribution, alphabetical ordered.

[1]Nevertheless, several studies have also documented limitations of SAEs (see Appendix P).

The second trade-off, which is largely overlooked in current literature, involves systematic vulnerability stemming from polysemantic structures in language models. In Anthropic's toy experiments, they note that stronger superposition can make models more vulnerable to adversarial attacks (Elhage et al., 2022). Beyond this, to our knowledge, there is very little existing empirical research that directly addresses the safety implications of polysemanticity in language models. In contrast, the vision model domain has a well-established body of work on various forms of adversarial model control that exploit polysemantic representations (Goh et al., 2021; Oikarinen & Weng, 2024a; Geirhos et al., 2023; Dreyer et al., 2024; Huang et al., 2022; Gorton & Lewis, 2025). Bereska and Gavves, in their review of mechanistic interpretability for AI safety, highlight polysemanticity as a key challenge in building safer LLMs (Bereska & Gavves, 2024). To bridge this gap, we focus on polysemantic structures in real-world LLMs, particularly those that persist across models, and explore hard-to-detect and targeted interventions to better understand their associated risks.

Before explaining the details, it is necessary to distinguish three nested representational domains:

**Human Symbolic Manifold** ($\mathcal{M}$): The latent first-order symbolic domain that encodes human-intuitive semantics independent of contextual usage.

**Model Activation Space** ($\mathcal{A}_\ell$): The $d$-dimensional vector space spanned by the neurons in layer $\ell$ of the language model; it partially reflects $\mathcal{M}$.

**Sparse Feature Basis** ($\mathcal{F}_\ell$): The $k$-dimensional, typically overcomplete basis ($k \gg d$) extracted from $\mathcal{A}_\ell$ by a SAE.

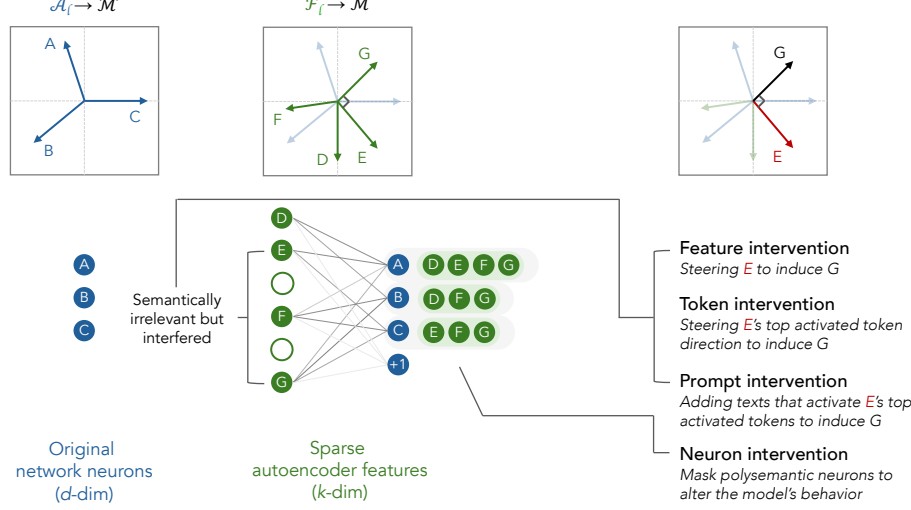

Figure 1: **Conceptual illustration.** (Top) Neurons A, B, and C span a 3D activation space $\mathcal{A}$, while D, E, F, and G denote SAE feature directions projected into the human symbolic manifold $\mathcal{M}$. In this example, features E and G are nearly orthogonal in $\mathcal{M}$ (i.e., semantically distinct), yet both load strongly on neuron C, so they still interfere in $\mathcal{A}$ via a shared activation direction. This illustrates that semantic orthogonality in $\mathcal{M}$ does not guarantee independence in activation space. (Bottom) Features are unevenly distributed across neurons: neuron A encodes more features than neurons B and C, forming a polysemantic "hub." Together, the panels highlight two vulnerabilities: (1) semantically distant features can interfere through shared activation geometry, and (2) polysemantic hubs concentrate interference risk on a small subset of neurons.

As illustrated in Figure 1, orthogonality in the activation space $\mathcal{A}_\ell$ does not persist after projection into the symbolic manifold $\mathcal{M}$. Consequently, two features from $\mathcal{F}_\ell$ that appear unrelated in $\mathcal{M}$ (i.e., anchoring semantically to distinct meanings under interpretation) can still interfere substantially in $\mathcal{A}_\ell$. This interference is also often unevenly distributed across neurons. Building on these two structural characteristics, we design feature, token, prompt, and neuron levels of intervention to investigate: (1) *how models' expression on a target feature is affected if semantically unrelated but interfering features are manipulated*, (2) *whether model vulnerability correlates with neuron polysemanticity, defined as the number of distinct features a neuron encodes*, and (3) *to what extent these polysemantic*

*interference patterns transfer across models*. In this work, model vulnerability to interventions is measured by the shift in the next-token prediction distribution following the intervention.

Our findings are four-fold. First, we present experimental evidence that interventions leveraging polysemantic structures of LLMs can effectively manipulate model outputs. Specifically, by targeting features and tokens — via steering vector techniques — and prompts — via prompt injection — that are not semantically aligned with the intended target but interfere with it, we can reliably induce the model to express the desired semantics. Second, we identify the existence of cross-model persistent polysemantic structures. By collecting shared interference features from both `Pythia-70M` and `GPT-2-Small` and applying them to steer `Llama-3.1-8B/70B-Instruct` and `Gemma-2-9B-Instruct`, we still observe substantial intervention effectiveness, revealing a consistent architecture of meaning that transcends specific implementations. Third, we explore those counterintuitive yet stable interference patterns that replicate across models. Post-hoc annotation of higher-order semantic relations accounts for a minority of cases, indicating that models learn robust regularities largely opaque to interpretation. Fourth, we analyze intervention at the neuron level and find that highly polysemantic neurons are more vulnerable: modifying their activation leads to greater semantic shifts in model output. For "super-neurons" (i.e., activated by over 500 features), amplification strongly alters model behavior, while deactivation has a notably reduced effect, suggesting they may serve as critical junctions in the semantic architecture.

## 2 PRELIMINARIES AND METHODS

### 2.1 SPARSE FEATURE EXTRACTION WITH SAEs

Our exploration of polysemantic structures draws on the pre-trained SAEs provided by *Neuronpedia*[2]. We focus on `GPT-2-Small` and `Pythia-70M`, the two models for which *Neuronpedia* supplies SAEs for all major sub-modules in every layer. Both SAEs have dimensionality 32,768, and we treat each SAE feature as defining an explicit direction in activation space.

Formally, let $f_{i,\ell}$ denote the $i$-th SAE feature associated with layer $\ell$, and let

$$d_{i,\ell} \in \mathcal{A}_\ell$$

be its projected direction in the model's activation space $\mathcal{A}_\ell$. We use the $\ell_2$-normalized directions

$$\hat{d}_{i,\ell} = \frac{d_{i,\ell}}{\|d_{i,\ell}\|_2}.$$

We define the *interference scale* between two SAE features $f_{i,\ell}$ and $f_{j,\ell}$ at layer $\ell$ as the cosine similarity between their activation-space directions:

$$\mathrm{I}_\ell(i,j) = \cos(\hat{d}_{i,\ell}, \hat{d}_{j,\ell}).$$

Analogously, let $m_i, m_j \in \mathcal{M}$ denote the corresponding directions of these features in the human symbolic manifold $\mathcal{M}$. Concretely, we approximate $\mathcal{M}$ using `text-embedding-3-large` embeddings of the natural-language glosses for features $i$ and $j$, generated by `GPT-4o-mini`, and normalize these vectors to unit length. We quantify their *surface-level semantic relatedness* by

$$\mathrm{S}(i,j) = \cos(m_i, m_j).$$

More details of SAEs are elaborated in Appendix D.

### 2.2 DISTINCT FEATURE IDENTIFICATION WITH AGGLOMERATIVE CLUSTERING

SAEs disentangle polysemantic neurons into monosemantic sparse features. These features, however, are not always decomposed at a consistent semantic level (Bricken et al., 2023; Foote, 2024). For example, a neuron associated with *dog*-related concepts might be divided into features representing different *dog breeds*, while another neuron encoding both *cat* and *car* concepts might be split into features representing *cat* and *car*. In such cases, the resulting monosemantic features differ in granularity. To mitigate this inconsistency, we employ agglomerative clustering to align feature

---

[2]`https://www.neuronpedia.org/`

representations to a consistent semantic level, facilitating both (1) the quantification of neuron polysemanticity and (2) the isolation of feature groups exhibiting low similarity in their surface meanings for subsequent analysis.

To identify distinct, higher-level features, we compute the semantic relatedness between pairs of SAE features using their auto-interpretation glosses (Caden Juang et al., 2024)[3] together with embeddings of the feature glosses. Prior work offers different heuristics for identifying semantically distinct SAE features. Foote et al. propose a cutoff of $0.5$ for distinguishing semantically distinct feature clusters (Foote, 2024), while another work on analyzing `text-embedding-3-large` shows that unrelated concepts typically fall within the $0.05$-$0.30$ cosine similarity range (Zuchen et al., 2025). Drawing on these insights, we conduct agglomerative clustering in each SAE layer using four increasingly strict similarity cutoffs (i.e., $0.40$, $0.30$, $0.20$, and $0.15$) to assess whether our results are robust across varying thresholds of semantic distinctness, while retaining sufficient feature density for experimentation. Figure 2 shows an example of the clustering results for the 5th MLP layer of `Pythia-70M` under the similarity cutoff of $0.4$. Detailed descriptive statistics on (1) the distribution of cosine similarities among SAE features, (2) the distribution of their interference values, and (3) the correlation between interference scale and semantic similarity are provided in Appendix F.

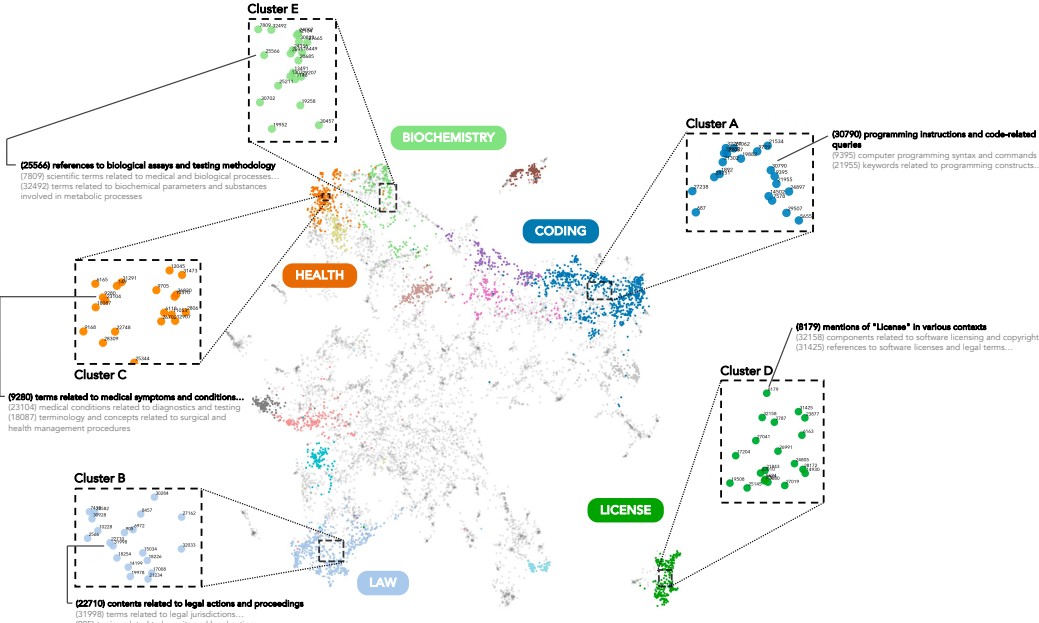

Figure 2: **Agglomerative clustering of SAE features trained on `Pythia-70M`'s 5th MLP layer under the cosine similarity threshold of $0.4$.** Only the five largest feature clusters are labeled.

## 2.3 EVALUATION CRITERIA

The effect of a polysemantic intervention is quantified as the change in alignment between the model's next-token prediction distribution and a target SAE feature $f \in \mathcal{F}_\ell$ on a context sentence. Details about sentence generation are elaborated in Appendix E. Specifically, we assess the similarity between the model's output and a feature-associated token set $T_f \subset V$, which consists of tokens with activation values above a threshold (here we use $0.8 \cdot$ highest activation value).

Let $O, \tilde{O} \in \Delta^{|V|}$ be the model's output distributions before and after intervention, and let $E \in \mathbb{R}^{|V| \times d}$ denote the token embedding matrix. Our main metric is **weighted cosine similarity**:

---

[3]Because different models can yield divergent auto-interpretations of the same feature, we conduct a cross-validation with `DeepSeek-V3` on a single SAE layer of `Pythia-70M`, which reveals both strong concordance between the interpretations produced by `GPT-4o-mini` and `DeepSeek-V3`. Replicating experiments with `DeepSeek-V3` feature glosses further confirm the consistency of our principal findings. See Appendix J.

$$c(O, T_f) = \sum_{t \in V} O(t) \cdot \max_{\bar{t} \in T_f} \cos(E_t, E_{\bar{t}}), \tag{1}$$

where each token $t \in V$ contributes its predicted probability weighted by the highest cosine similarity between its embedding and those in $T_f$. This metric aims at capturing the overall semantic alignment between model's output and the target feature. Then, the intervention effect is:

$$\Delta c = \frac{c(\tilde{O}, T_f) - c(O, T_f)}{c(O, T_f)}. \tag{2}$$

As an alternative, we also report **weighted overlap**, which simply sums the output probability mass over $T_f$. Formal definition and results are provided in Appendix G. In comparison, this metric captures how the model's output is steered towards the most related tokens of the target feature.

## 2.4 Overview of Intervention Methods

Our investigation of polysemantic interventions begins with `Pythia-70M` and `GPT-2-Small`, using three complementary approaches: **feature-direction steering**, **token-gradient steering**, and **prompt injection**. We randomly select target features to be intervened[4]. For each selected target feature, we sample interference features from feature clusters derived from Section 2.2—excluding the target's own cluster—to ensure sufficient meaning dissimilarity with the target. Interference features are drawn from five interference intervals: $[0.0, 0.1]$, $[0.1, 0.2]$, $[0.2, 0.3]$, $[0.3, 0.4]$, and $[0.4, 1.0]$. In the first two experiments, we construct steering vectors for interference features using two methods: (1) projecting feature directions from the sparse basis $\mathcal{F}_\ell$ into the model's activation space $\mathcal{A}_\ell$, and (2) computing token-gradient directions from the partial derivatives of the layer's activations with respect to each feature's top-activating tokens. For each intervention, we roughly optimize the scaling of the steering vector over the range $[-20, 20]$, balancing intervention strength with the need to preserve coherent model outputs (i.e., avoiding substantial disruption to the overall output distribution). Details about the tuning strategy are elaborated in the Appendix I.2. In the prompt injection setting, we prepend sampled snippets of top-activating texts from interference features to the input and measure the semantic shifts of the model's output using metrics stated above, conditioned on varying levels of feature interference.

In the second step, we incorporate an evaluation on the cross-model transferability of polysemantic structures. Particularly, we apply the two scalable intervention methods (i.e., token-gradient steering and prompt-injection) to black-box, larger models `Llama-3.1-8B/70B-Instruct` and `Gemma-2-9B-Instruct` to see whether same effects manifest. For the steering intervention, we select target features and sample interference features that are consistently identified in both `Pythia-70M` and `GPT-2-Small`. We then extract the corresponding gradient vectors and use them to steer the black-box models. For the prompting intervention, we prepend activation-inducing text snippets shared across the identified interference features to the input prompts. One might question the practical significance of our next-token control test. While the primary objective of this study is mechanistic, we additionally conduct a small-scale evaluation of the gradient-based intervention—which we empirically show stronger interference effects—on the *HellaSwag* dataset. This evaluation assesses whether the intervention can steer model behavior in target-related contexts without degrading overall task performance. Implementation details are provided in Appendix L.

Finally, we analyze the impact of **neuron polysemanticity** on model outputs in `Pythia-70M` and `GPT-2-Small`. For each neuron, we identify strongly connected features by thresholding connection weights at $0.2$, and define its degree of polysemanticity as the number of such features. We then suppress or amplify the activations of neurons with varying polysemanticity levels and evaluate how the model's output distribution shifts toward the semantics of associated feature clusters.

---

[4]We randomly sample 480 target features from `GPT-2-Small` and 180 from `Pythia-70M` (See Appendix H for details).

## 3 EXPERIMENTS

### 3.1 EXPLOITING SAE FEATURE DIRECTIONS FOR INTERVENTION

Our hypothesis posits that if two feature directions interfere in $\mathcal{A}_\ell$, despite being nearly orthogonal in $\mathcal{M}$, then enhancing one will inevitably influence the other to some degree. If true, this would imply the potential to covertly manipulate the output probability of a target feature by steering with not obviously related features. To evaluate this, we pair each target feature with interference features sampled at varying levels of semantic dissimilarity. We vary feature irrelevance thresholds (from $0.4$ to $0.15$; Appendix H) to confirm that the observed pattern holds across threshold choices. Additionally, we perform controlled regressions—accounting for feature-pair cosine similarity—under two linear model specifications to confirm that interference effects persist independently of semantic similarity. Figure 3 reports the results, and Table 3 in Appendix H shows particular examples.

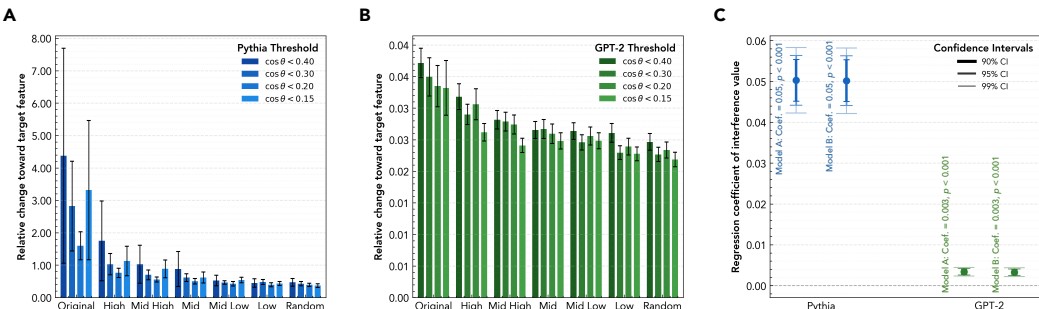

Figure 3: **Interventions along interfering but semantically distinct SAE feature directions reliably steer next-token predictions toward the desired semantic direction**. (A–B) Relative change in weighted cosine similarity toward the target ($\Delta c$). Bars show the mean relative change compared to baseline across interference levels, with lighter shades indicating stricter feature-meaning relevancy cutoff thresholds. The x-axis denotes the interference scale between the target and intervention feature: *Original* corresponds to intervening with the target feature itself, and *Random* serves as a random feature intervention baseline. Error bars denote 95% confidence intervals. (C) Regression estimates of the effect of feature-pair interference value on intervention success. Two regression specifications are shown: Model A regresses weighted cosine similarity after intervention ($c(\tilde{O}, T_f)$) on interference value, with feature-meaning similarity, baseline weighted cosine similarity ($c(O, T_f)$), and layer-type controls; Model B regresses the change score ($\Delta c$) on interference value, with feature-meaning similarity and layer-type controls. Error bars denote 90%, 95%, and 99% confidence intervals. Results with the alternative metric are shown in Figure 10.

All analyses consistently show that steering with features that are semantically dissimilar yet interfering can alter the output probabilities of a target feature's top-activating tokens, with stronger effects observed at higher interference levels. We also find that SAE-based interventions are generally much less effective on `GPT-2-Small` than on `Pythia-70M`. An intuitive hypothesis is that greater model depth attenuates the downstream impact of mid-layer activation shifts (Fort, 2023). To probe this, we run an additional SAE-steering experiment on a third model, `Gemma-2-2B`, on a smaller evaluation set. Despite being deeper than `GPT-2-Small`, `Gemma-2-2B` exhibits a larger effect under weighted cosine similarity than `GPT-2-Small`, though still a smaller effect than `Pythia-70M` (full discussion in Appendix O). At the same time, its gains under the weighted overlap metric are comparable to those of the two smaller models. Taken together, these results suggest that model depth alone cannot account for the differences in cosine-based effect sizes; at least part of the discrepancy appears to arise from how different models interact with our two evaluation metrics.

### 3.2 STEERING WITH GRADIENT VECTOR FOR TOKEN INTERVENTION

In this section, we treat the top-activating tokens of semantically unrelated yet interfering SAE features as intervention signals (Ferrando et al., 2024). For each interference feature, we pass its top-activating text through the model and, at the feature's corresponding layer, compute the gradient of that token with respect to all neurons in that layer. This gradient serves as the steering vector.

As shown in Figure 4, using token-gradient steering on both models yields roughly $\sim 10\times$ larger effects than steering along SAE feature directions. Interestingly, token-gradient steering also flattens the relationship between interference scale and intervention effectiveness, and steering the original feature's gradient direction is less effective than steering the interfering ones. This may stem from the fact that token gradients are not tied to SAE-defined interference levels and SAE features can exhibit a degree of arbitrariness (Paulo & Belrose, 2025; Heap et al., 2025).

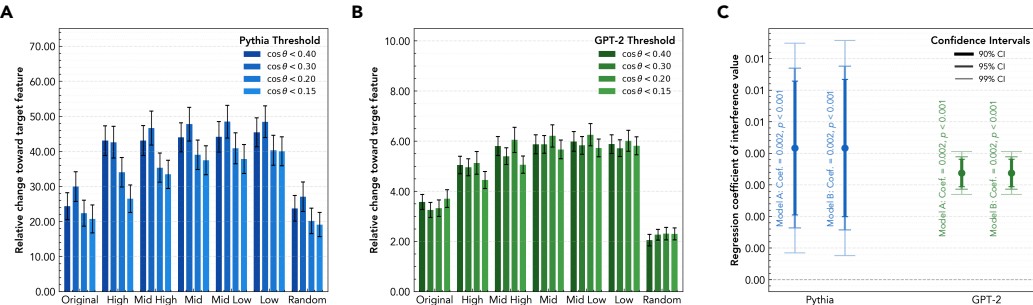

Figure 4: **Interventions along gradient directions of top-activating tokens for interfering but semantically distinct SAE features reliably steer next-token predictions toward the desired semantics.** Subpanels follow the same conventions as Figure 3, but intervention vectors are computed from token gradients rather than SAE decoder weights. For (A–B), error bars indicate 95% confidence intervals; for (C), error bars denote 90%, 95%, and 99% confidence intervals. Results with the alternative metric are shown in Figure 11.

### 3.3 PROMPT INJECTION FOR INFERENCE TIME INTERVENTION

In addition to the two intervention methods described above, which directly modify the model's internal activations, interference effects may also arise from prompt injection. In this experiment, we still employ the same method as the previous two experiments to sample target and interference features. The difference is that we extract continuous text snippets with high activation values from the activation text of the interference features and prepend them to the prompts. This is intended to concisely write the semantics of the interference features into the model's latent space, while avoiding disrupting the semantics of the prompts, compared to pasting the complete activation text or pasting it within or after the prompts.

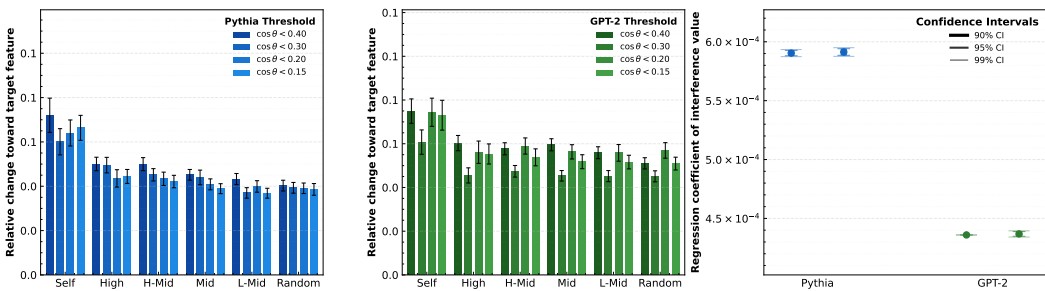

Figure 5: **Prepending highly-activating tokens of the interference features to prompts moderately shift next-token predictions toward the desired semantics.** Subpanels follow the same conventions as Figure 3, while intervention arise from snippets of highly-activating texts of the interference feature. For (A–B), error bars indicate 95% confidence intervals; for (C), error bars denote 90%, 95%, and 99% confidence intervals. Results with the alternative metric are shown in Figure 15.

The overall interference magnitude—and the monotonic increase in semantic shift under stronger interventions (Figure 5)—is weaker than that achieved by direct latent-space manipulation, at roughly the 10% level. However, when evaluated using weighted overlap, the pattern becomes pronounced: intervention effects increase by approximately $10\times$–$100\times$ in `GPT-2-Small` and $100\times$–$1000\times$ in

`Pythia-70M`. This suggests that prompt injection concentrates interference on a small subset of tokens most strongly associated with the target feature.

## 3.4 Generalization of Polysemantic Intervention Vulnerability

Similar polysemantic interference may exist across multiple models. Note that interfering by token gradient vector and prompt injection do not rely on SAEs, they can be applied to models without internal access. We therefore target `Llama-3.1-8B/70B-Instruct` and `Gemma-2-9B-Instruct`, selecting several target feature types which widely appear in SAE database provided by *Neuronpedia*, including location, name, number, science, emotion, animal, color and time. Their interference features in two small models `GPT-2-Small` and `Pythia-70M` are viewed as their potential interference features in other models.

For gradient vector-based interventions, we target at `Llama-3.1-8B-Instruct` and `Gemma-2-9B-Instruct`, we select highly-activating tokens from the activation texts of the interference features shared by the two small models and extract the steering vectors from them. Through fine-tuning of the steering strength, we could notably boost the presence of relevant tokens in the top-10 prediction list with over 95% success rate in feature types such as location, name and emotion, as shown in Appendix K. Prompt injection interventions are tested on `Llama-3.1-8B/70B-Instruct` and `Gemma-2-9B-Instruct`. As shown in Table 1, for some feature types, high-interference tokens derived from the two small models can steer larger models more effectively than random baselines. However, for several other feature types, the generalizability of their interference structures is not salient, as shown in Table 5.

Notably, we find some specific counterintuitive interference pairs, such as location and datatypes in programming languages as shown in Table 6, that exist across more than two models. In hindsight, these results suggest that shared polysemantic structures observed in small models also extend to larger models, indicating generalized vulnerabilities that persist across architectures and training regimes.

Table 1: **Comparing intervention effect of prompt injection**

| Target | Model | Original | High-interference | Low-interference | Random |
|---|---|---|---|---|---|
| Locations | Pythia-70M | 65.08%*** | 36.93%** | 32.53% | 35.06% |
| | GPT-2-Small | 44.68%*** | 18.42%*** | 19.08%*** | 16.42% |
| | Llama-3.1-8B-Instruct | 33.84%*** | 20.78%*** | 19.63%* | 18.24% |
| | Gemma-2-9B-Instruct | 10.16% | 12.71%*** | 11.36% | 10.66% |
| | Llama-3.1-70B-Instruct | 37.23%*** | 28.21%*** | 23.09%** | 24.48% |
| Number | Pythia-70M | 65.32%*** | 35.15% | 36.71%*** | 34.30% |
| | GPT-2-Small | 55.87%*** | 30.33% | 31.23% | 34.71% |
| | Llama-3.1-8B-Instruct | 55.97%*** | 31.87%* | 32.90%*** | 30.57% |
| | Gemma-2-9B-Instruct | 48.42%*** | 29.93%*** | 30.64%*** | 27.16% |
| | Llama-3.1-70B-Instruct | 25.57%*** | 8.85%*** | 6.67% | 7.09% |
| Science | Pythia-70M | 61.66%*** | 23.13% | 22.78% | 28.58% |
| | GPT-2-Small | 75.70%*** | 25.93%*** | 26.07%*** | 21.71% |
| | Llama-3.1-8B-Instruct | 49.67%*** | 20.08%*** | 18.95% | 17.94% |
| | Gemma-2-9B-Instruct | 46.84%*** | 20.40% | 19.25% | 20.15% |
| | Llama-3.1-70B-Instruct | 67.26%*** | 48.22%*** | 43.57% | 42.24% |

*Note*: Cell values show the success rate of elevating target-type tokens into the top-30 predictions. Gray-shaded rows indicate black-box interventions. Testing uses a shared token set from the two small models. ***, **, and * denote t-test significance at $p < 0.001$, $p < 0.01$, and $p < 0.05$, respectively, vs. random baseline. High- and low-interference tokens lie in $[0.5, 1.0]$ and $[0.2, 0.5]$ for `Pythia-70M`, while $[0.3, 1.0]$ and $[0.2, 0.3]$ in `GPT-2-Small`. Details in Appendix K.2.

## 3.5 Analyzing Shared but Counterintuitive Polysemantic Structure

We observe that many similar feature pairs in `GPT-2-Small` and `Pythia-70M` that lie far apart in the first-order symbolic manifold, $\mathcal{M}$ (by surface/gloss semantics), nevertheless lie close in both models' activation space $\mathcal{F}$. To understand this transferability, we consider mechanisms that couple

features via higher-order semantic structure, including semantic-priming-type associations (e.g., thematic/scripts, causal, frame roles) and morphological relatedness, even when overt meanings appear unrelated (Mandera et al., 2017; Bojanowski et al., 2017).

To probe these links, we conduct large-scale annotation with `DeepSeek-V3` and `GPT-5-mini`. Models are prompted to identify higher-order semantic relations for shared interfering feature pairs that satisfy three criteria: interference $> 0.4$, semantic similarity in $\mathcal{M} < 0.2$, and cross-model feature-pair semantic similarity $> 0.5$. In total, 459,229 pairs are annotated (prompts, rubric, and head-to-head annotator comparison in Appendix M). Only 27.7% of pairs are judged by at least one model to exhibit a plausible higher-order association, and most such links remain counterintuitive upon post-hoc inspection. Because the association-check task can be challenging for LLMs, we also run a conservative paired-choice evaluation. We sample 3,800 high-interference, low-similarity pairs. For each, we randomly draw a comparison pair strictly matched on semantic similarity but with substantially lower interference, then ask `GPT-5-mini` to select which pair is more related (details in Appendix M). The high-interference pair is chosen 64.3% of the time (Wilson 95% CI [0.628, 0.658]); a one-sided exact binomial test against 0.5 is significant ($p < 0.001$), corresponding to a log-odds of 0.589 in favor of the high-interference pair, indicating a statistically meaningful yet modest effect. Taken together, these results both validate our filtering strategy for isolating unrelated pairs in the intervention analyses and point to a striking regularity: *LLMs instantiate stable, cross-model polysemantic organization that is often opaque to semantic intuition.*

In Appendix M, Table 8 reports examples where models detect latent associations between feature pairs and where they do not. One notable case is the last, asterisked example in Table 8: both annotators label it as negative. In our follow-up analysis, however, we hypothesize a biographical–affective link: mentions of "Beethoven" may co-occur with expressions of frustration/suffering, given his late-life deafness and celebrated late-period compositions. These examples suggest that LLM polysemanticity may approximate latent knowledge structures and offer testable hypotheses. In summary, it is possible that human studies could reveal comparable human recognition of the same "weak signals" picked up across models, even if they cannot recall or justify them. A full adjudication of this possibility lies beyond our present scope and we leave for future work.

### 3.6 MANIPULATING ACTIVATIONS FOR NEURON INTERVENTION

To complete our discussion, we explore models' vulnerability to interventions on individual neurons. Specifically, we investigate how the degree of polysemanticity in neurons affects the output. For aggregated features obtained through agglomerative clustering under the cosine similarity threshold of 0.4, we quantify each neuron's connected number of features. Here, we only involve neuron-feature pairs with a connection strength greater than 0.2. Among all neurons, those connected to only one or two aggregated features account for more than 33% in strongly connected neurons, as shown in Figure 17. In addition to neurons connected to multiple or dozens of aggregate features, there are also some "super-neurons" with connections exceeding 500. We examine the impact of manipulating these neurons on the model's output. Experimental results, as shown in Figure 6, indicate that neurons with higher degrees of polysemanticity are more vulnerable, which means they tend to affect model outputs more effectively. For certain "super-neurons," however, the impact on the model is notably asymmetric: masking them results in less influence than neurons with lower polysemanticity, while amplifying their activations often leads to exponentially greater effects on model behavior.

## 4 DISCUSSION

This work makes two contributions. First, we systematically investigate the vulnerability of LLMs to structured interventions grounded in their polysemantic representations. Specifically, we examine three types of intervention: (1) **feature direction-based**, (2) **token gradient-based**, and (3) **prompt-based**. Feature direction interventions rely on SAEs. While less effective than gradient-based approaches, they form the basis for deriving token gradient vectors. Token gradient-based interventions are most effective and can be constructed directly from activation texts, without requiring SAE pre-training—although they assume access to internal activations. Prompt-based interventions require minimal access and, despite their surface-level nature, still yield meaningful behavioral shifts. Additionally, we explore **neuron-level interventions**, motivated by the uneven distribution of features across neurons. We find that the behavioral impact of masking and amplification correlates

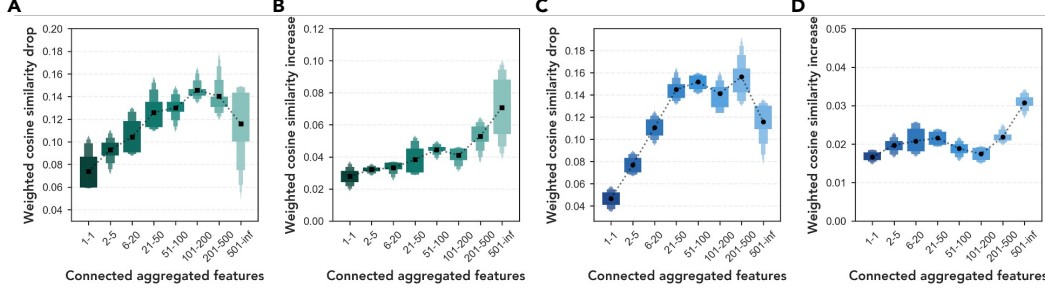

Figure 6: **Effects of neuron activation and suppression on model behavior depend on neuron polysemanticity level.** The $x$-axis indicates neuron categories grouped by the number of connected features (after clustering). The $y$-axis reports the change in weighted cosine similarity. Each box plot centers on the median ($50\%$) and progressively splits the remaining data in half at each level. A and B correspond to `GPT-2-Small`; C and D correspond to `Pythia-70M`. A and C show the effect of masking neuron activations, while B and D show the effect of amplifying them.

with neuron polysemanticity. We also identify a class of "super-neurons," those encoding over $500$ features, for which amplification significantly alters model behavior, while deactivation results in a markedly reduced effect.

The second contribution lies in our finding that polysemantic, structural vulnerabilities identified in two small models transfer to larger, instruction-tuned black-box models (e.g., `Llama-3.1-8B/70B-Instruct`, `Gemma-2-9B-Instruct`) via token-gradient- and prompt-level manipulations, producing predictable behavioral shifts. This suggests that certain polysemantic structures are preserved across architectures and training regimes, exposing a shared representational basis. This directly challenges prevailing theories that treat polysemanticity as an incidental artifact of training (Marshall & Kirchner, 2024; Lecomte et al., 2023). Our exploratory analyses further suggest that these transferable polysemantic structures are not reducible to higher-order relations readily intelligible to humans; we therefore treat these counterintuitive regularities as testable hypotheses about latent knowledge structure. These results sharpen a central question about LLM polysemanticity: are they unintended byproducts or stable, higher-order patterns that await rigorous examination? Our results point towards the latter. Finally, our findings strengthen recent evidence of representational consistency and topological stability across models (Huh et al., 2024; Wolfram & Schein, 2025; Lee et al., 2025), even as the origins and functional implications of this consistency remain open challenges. Our work is the first to systematically evaluate polysemantic structural vulnerabilities in real-world LLMs, but it has several limitations, including intervention depth and transferability robustness tests. We discuss these limitations and ethical considerations in Appendix P.

## 5 CONCLUSION

We systematically probe the vulnerability of LLMs to structured interventions grounded in the polysemantic representations of two small models using SAEs. We show that model behavior can be steered toward specific feature directions by manipulating semantically unrelated yet interfering features via three intervention methods. Interventions distilled from polysemantic structures shared across the small models transfer to larger, black-box instruction-tuned models, indicating a stable and transferable polysemantic topology that persists across architectures and training regimes. Post-hoc annotation suggests that fewer than $30\%$ of these shared interference structures align with higher-order relations readily intelligible to humans; many counterintuitive cases may therefore serve as generators of testable hypotheses about latent knowledge structure. Finally, by leveraging the uneven distribution of features across neurons, we assess models' sensitivity to neuron-level manipulations across degrees of polysemanticity and reveal asymmetric effects in "super-neurons." Together, these findings provide a foundation for future work on the structural properties, vulnerabilities, and representational robustness of LLMs.

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

# CONTENT OF APPENDIX

# A    RELATED WORK

## A.1    A BRIEF REVIEW ON LLM ADVERSARIAL INTERVENTIONS

Over the past five years, a growing body of high-impact research has revealed that even aligned LLMs remain vulnerable to a set of converging attack strategies. First, *prompt-space jailbreaks* have evolved from handcrafted exploits into automated, highly transferable methods. For instance, a single gradient-and-greedy–optimized "universal suffix" can consistently bypass refusal policies in ChatGPT, Bard, Claude, and a wide range of open-source models—demonstrating both query efficiency and cross-model generalizability (Zou et al., 2023). Second, *activation-space steering* techniques like Contrastive Activation Addition (CAA) show that simple linear interventions in the residual stream can steer behaviors such as hallucination, sycophancy, or toxicity with minimal performance degradation (Panickssery et al., 2023). Third, *parameter-space backdoors*, such as the Composite Backdoor Attack, embed stealthy triggers during fine-tuning that achieve near-perfect malicious compliance without affecting standard benchmarks (Huang et al., 2023). Mechanistic interpretability offers a unifying explanation: transformer activations encode more features than they have dimensions, forcing representations into a compressed superposition and leading to widespread polysemantic overlap (Elhage et al., 2022). Recent work with SAEs has begun to isolate—and in some cases manipulate—these overlapping features directly (Nanda, 2024). Building on this insight, our intervention targets SAE-derived polysemantic directions, integrating prompt-, activation-, and neuron-level interventions into a unified, transferable framework that broadens the known landscape of LLM vulnerabilities.

## A.2    A BRIEF REVIEW ON SAE-BASED INTERVENTION TECHNIQUES IN LLMS

SAE-based interventions represent a promising direction for developing more interpretable and controllable LLMs. Recent research have introduced a diverse set of SAE-based techniques, such as clamping, patching, and causal tracing, applied across a range of use cases (Farrell et al., 2024; Cunningham et al., 2023; Marks et al., 2024). Empirical results indicate that these methods can be highly effective. For example, targeted unlearning via SAE features has been shown to suppress undesired capabilities with fewer side effects than global fine-tuning (Khoriaty et al., 2025; Muhamed et al., 2025), while feature-level steering enables more nuanced output control than prompt-based methods alone (Rajamanoharan et al., 2024). A key advantage of SAE-based approaches is their efficiency at inference time: they often require only a forward pass with lightweight vector operations and typically do not require model retraining, making them well-suited for real-time interventions.

However, the approach is still in its early stages. Key limitations include challenges in achieving complete and disentangled feature representations, which depend heavily on SAE training quality and selection procedures (Chanin et al., 2024). Computational overhead remains non-trivial, though recent developments such as $k$-sparse autoencoders and JumpReLU activations offer promising improvements in scalability (Rajamanoharan et al., 2024). There is also a growing need for standardized evaluation benchmarks tailored to intervention methods. A unified benchmark would enable more meaningful comparisons across studies. Currently, researchers often rely on custom evaluation protocols, limiting cross-paper comparability.

In summary, SAE-based interventions offer a powerful mechanism for both understanding and steering model behavior. They uniquely bridge interpretability and utility: not only can we decode model activations into human-interpretable concepts (Cunningham et al., 2023), but we can also use those same features to drive controlled behavioral change (Khoriaty et al., 2025). In this work, rather than focusing on a specific downstream application, we leverage SAEs to investigate structural sensitivities in LLMs—demonstrating that polysemantic features can serve as a substrate for transferable, interpretable interventions. This perspective highlights the broader role of SAEs in the design of more transparent and controllable AI systems.

## B  IMPACT STATEMENT

This work systematically investigates a semantic vulnerability in LLMs rooted in polysemanticity—where single neurons encode multiple semantically dissimilar features. We introduce four complementary approaches that expose this vulnerability: manipulating SAE-derived features, token gradients, and prompts to steer model outputs via semantically unrelated inputs, and intervening at the neuron level to reveal a correlation between polysemanticity and output sensitivity. We also identify a class of "super-neurons" whose amplification disproportionately alters model behavior, while masking them has a limited effect. These findings not only highlight the unique characteristics of the structural fragility of LLMs but also provide practical tools for probing and controlling their internal mechanisms. Our work lays a foundation for future research in AI safety and mechanistic interpretability, not only enabling defenses against such vulnerabilities and more targeted interventions for alignment, but also offering a theoretical lens into the model's internal organization, revealing stable yet counterintuitive interference patterns that may reflect a form of unconscious knowledge association.

## C  THE USE OF LARGE LANGUAGE MODELS (LLMs)

LLMs play a significant role in our study. We list the usages of LLMs below.

**Synthetic data generation.**  For the dataset used for intervention experiments, we use `DeepSeek-V3` to generate incomplete sentences for next-token completion. Details are elaborated in Appendix E.

**SAE feature auto-interpretation.**  The SAE feature glosses in the main experiments are generated by `GPT-4-mini`. For cross-validation, we also annotate a subsample of SAE features using `DeepSeek-V3`. Details in Appendix J.

**Feature pair association analysis.**  To investigate whether the seemingly unrelated interference feature pairs present higher-order associations that are still comprehensible, we ask `GPT-5-mini` and `DeepSeek-V3` to label these feature pairs for a post-hoc check. Details in Appendix M.

**Grammar check for paper writing.**  We use LLMs to refine phrasing, correct grammar, and improve readability during manuscript preparation, while all substantive ideas, analyses, and conclusions remain our own.

## D    SPARSE AUTOENCODER TRAINING

SAEs are a rapidly developing tool for probing the polysemantic structure of neurons (Shu et al., 2025). Given the activation vector $\mathbf{a} \in \mathbb{R}^{d_{\text{embed}}}$ from a particular model layer, an SAE projects it into a higher-dimensional sparse code $\mathbf{f} \in \mathbb{R}^{d_{\text{sae}}}$ in order to disentangle the multiple semantics that a single neuron may simultaneously encode. The forward computation and the resulting feature definition $\mathbf{f}$ are shown below:

$$\mathbf{f} = \text{Act}\big(W_{\text{enc}}\mathbf{a} + \mathbf{b}_{\text{enc}}\big),$$

$$\bar{\mathbf{a}} = W_{\text{dec}}\mathbf{f} + \mathbf{b}_{\text{dec}}.$$

The encoder and decoder parameters are

$$W_{\text{enc}} \in \mathbb{R}^{d_{\text{sae}} \times d_{\text{embed}}}, \quad W_{\text{dec}} \in \mathbb{R}^{d_{\text{embed}} \times d_{\text{sae}}}, \quad \mathbf{b}_{\text{enc}} \in \mathbb{R}^{d_{\text{sae}}}, \quad \mathbf{b}_{\text{dec}} \in \mathbb{R}^{d_{\text{embed}}}.$$

where *Act* is an activation function, such as ReLU and TopK for `Pythia-70M` and `GPT-2-Small`. The SAE is trained by dictionary learning to minimize

$$\mathcal{L} \;=\; \big\|\mathbf{a} - \bar{\mathbf{a}}\big\|_2^2 \;+\; \lambda \sum_i \mathbf{f_i}\,\big\|\mathbf{W}_{\text{dec}\,[\cdot,i]}\big\|_2,$$

where the first term is the reconstruction loss and the second encourages sparsity (weighted by $\lambda$).

For each feature $f_i$, its direction in the embedding space is defined as the *unit-norm* decoder column.

$$\hat{\mathbf{d}}_i \;=\; \frac{W_{\text{dec}\,[\cdot,i]}}{\big\|W_{\text{dec}\,[\cdot,i]}\big\|_2}.$$

# E   DATASET GENERATION

The investigation of the interference effects of SAE features is conducted within the contexts where they are likely to fire, which allows us to manipulate the expression of these features without significantly compromising the model. For example, when given the prompt "In the weekend, we are going to", the features related to locations are incorporated into our examination. Additionally, we observe a positive correlation between the boost of a SAE feature and the increased probability of its high-activation tokens in the model's next-token predictions. Therefore, here we simply construct sentences for each token in the model's vocabulary, ensuring that these sentences are grammatically capable of leading to the token and evaluate the expression of features with respect to the tokens' output probabilities.

**System:** Generate exactly 3 incomplete English sentences where the next word would clearly be "target_token".  Return a JSON dictionary where:
- The ONLY key is the exact "target_token" (including spaces/capitalization) – The value is a list of 3 sentence fragments that naturally lead to "target_token"
Example for "target_token=' apple'":
{
" apple":  [
"She reached into the basket and grabbed",
"The teacher pointed to the red",
"He washed and polished his"
]
}
Rules:
1.  All sentences MUST grammatically require "target_token" next to it
2.  Use different contexts / scenarios for variety 3.  Maintain exact formatting – no additional keys or explanations
**User:** target_token={token}

We also use `DeepSeek-V3` to roughly classify the token types:

**System:**  You are a linguistic analyzer.  Your task is to classify tokens from language model vocabularies into semantic categories.

Given a list of tokens, classify each token into ONE of these categories:

- person:  Names, pronouns, occupations, human-related terms
Examples:  " John", " Mary", " doctor", " teacher", " he", " she", " people"

- location:  Cities, countries, geographical features, places
Examples:  " London", " America", " mountain", " beach", " city", " Paris", " China"

- time:  Time units, dates, temporal expressions, seasons
Examples:  " Monday", " January", " morning", " year", " day", " week", " winter"

- number:  Numerical digits, number words, mathematical terms
Examples:  " one", " two", " 1", " 2", " first", " hundred", " plus", " minus"

- animal:  All living creatures, insects, pets, wildlife
Examples:  " dog", " cat", " bird", " lion", " fish", " elephant", " butterfly"

- food:  Edible items, drinks, cooking terms, ingredients
Examples:  " apple", " bread", " water", " coffee", " chicken", " rice", " pizza"

– color:  Color names, shades, visual descriptors
Examples:  " red", " blue", " green", " black", " white", " yellow", "
purple"

– emotion:  Feelings, emotional states, psychological terms
Examples:  " happy", " sad", " angry", " love", " fear", " excited", "
worried"

– body:  Human/animal body parts, anatomy
Examples:  " head", " hand", " eye", " heart", " leg", " face", " brain"

– transport:  Vehicles, transportation methods
Examples:  " car", " bus", " train", " plane", " ship", " bicycle", "
truck"

– science:  Scientific terms, elements, physics/chemistry concepts
Examples:  " oxygen", " energy", " atom", " DNA", " gravity", " electron",
" acid"

– abstract:  Philosophy, ideas, concepts, mental constructs
Examples:  " freedom", " justice", " truth", " idea", " concept", "
theory", " belief"

– object:  Physical items, tools, furniture, equipment
Examples:  " table", " chair", " book", " phone", " computer", " tool", "
box"

– action:  Verbs, activities, movements, processes
Examples:  " run", " walk", " eat", " think", " write", " jump", " sleep"

– unknown:  Anything that doesn't clearly fit into the above categories

Return a JSON object where each key is a token and each value is its
category.

Rules:
1.  Classify ALL provided tokens
2.  Use EXACT token strings as keys (including spaces)
3.  Choose the MOST appropriate single category
4.  Use lowercase category names only
5.  When in doubt, use "unknown"
6.  Return ONLY the JSON object, no additional text

**User:** Classify these tokens:
token_str_1, token_str_2, ....

Here we provide a table of some example tokens and their sentences generated by `DeepSeek-V3` (See Table 2).

Table 2: **Token type and prompt sentences examples**

| Token | Type | Sentence Examples |
|---|---|---|
| London | location | *After a long flight, we finally arrived in __*
*The train from Paris was heading straight to __*
*She always dreamed of visiting the historic city of __* |
| harbor | location | *The cruise ship slowly approached the bustling __*
*Fishermen gathered at the edge of the protected __*
*The city's economy thrived thanks to its busy __* |
| Mike | person | *After the meeting, everyone turned to __*
*The teacher called on __*
*She handed the report directly to __* |
| Trump | person | *The media has been closely following the latest statements from __*
*During the debate, the moderator asked a direct question to __*
*Many supporters gathered outside the venue to catch a glimpse of __* |
| expert | person | *After years of practice, she became an __*
*The company hired an __*
*When it comes to antique furniture, he's an __* |
| loves | emotion | *She truly believes that everyone __*
*The way he looks at her shows how much he __*
*Despite their differences, their friendship __* |
| hates | emotion | *Everyone knows that she __*
*The way he treats people shows he __*
*It's clear from his expression that he __* |
| apple | object | *She reached into the bag and pulled out __*
*The smoothie recipe called for one chopped __*
*He carefully balanced the shiny red __* |
| sad | emotion | *After hearing the bad news, she felt incredibly __*
*The movie's ending left everyone feeling __*
*His eyes told a story of being deeply __* |
| happy | emotion | *After receiving the good news, she felt extremely __*
*The children were laughing and playing, clearly very __*
*Winning the competition made him incredibly __* |

# F SUPPORTIVE STATISTICS

Active features refer to SAE features that have input texts enabling them to reach an active state. In addition to the semantic clustering of active features mentioned in the main text, we also apply agglomerative clustering to cluster their interference values. The threshold for dividing clusters is set to $0.4$. As shown in Figure 8, the vast majority of clusters contain only one feature, indicating that only a small number of features exhibit high interference with others.

In Figure 9, we also provide an overview of the distribution of semantic similarity and interference values between features, as well as their correlations. Since the distribution and correlation of these two types of data remain largely consistent across all layers of the model, we present a representative example layer for illustration.

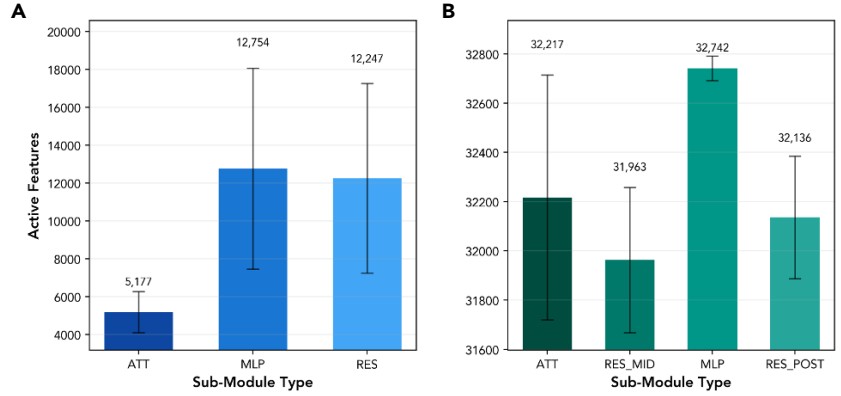

Figure 7: **Number of active features extracted by SAEs per layer.** A is the result of `Pythia-70M`, and B is the result of `GPT-2-Small`. Error bars represent $95\%$ confidence intervals.

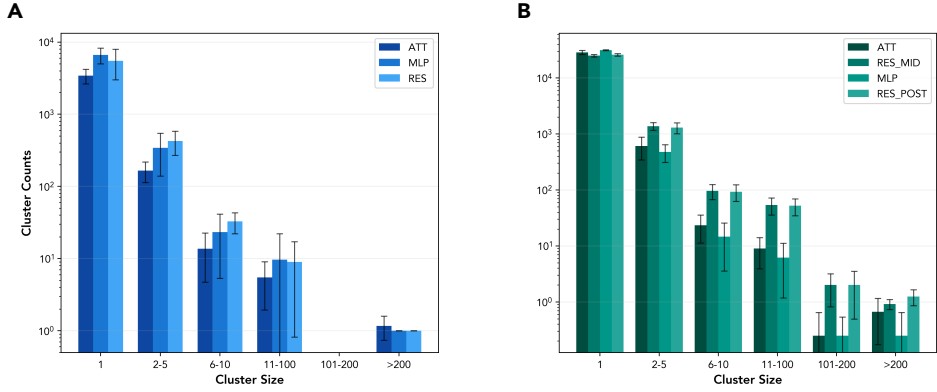

Figure 8: **Interference Cluster Size Distribution.** A is the result of `Pythia-70M`, and B is the result of `GPT-2-Small`. Error bars represent $95\%$ confidence intervals.

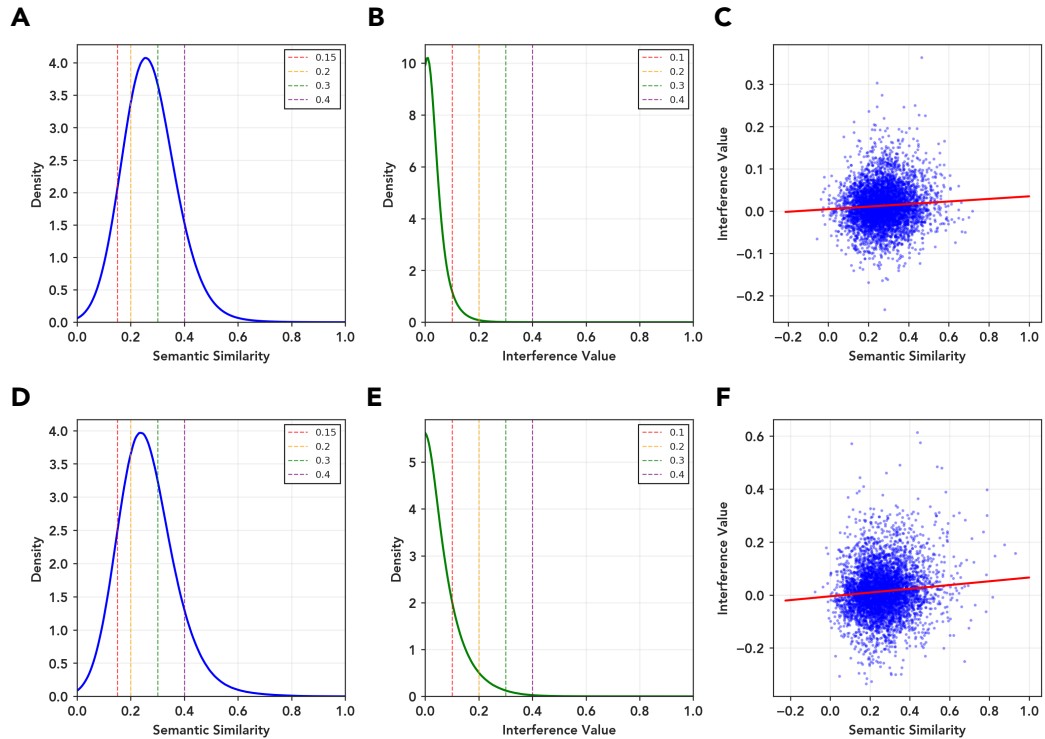

Figure 9: **Semantic similarity and interference distribution of `Pythia Res-4` and `GPT2-Small Res Post-4` layer.** (A–C) are the results of `Pythia-Res-4`. For semantic similarity, $14.0\%$ of values are below 0.15, $29.6\%$ are below 0.2, $67.4\%$ are below 0.3, and $89.6\%$ are below 0.4. For interference, $85.5\%$ of values are below 0.1, $96.4\%$ are below 0.2, $99.2\%$ are below 0.3, and $99.8\%$ are below 0.4. The bivarite analysis suggests semantic similarity and interference value is positively associated ($\beta = 0.071$, $s.d = 0.092$, $p < 0.001$). (D–F) are the results of `GPT-2-Smal Res Post-4`. For semantic similarity, $10.8\%$ of values are below 0.15, $24.5\%$ are below 0.2, $63.3\%$ are below 0.3, and $89.5\%$ are below 0.4. For interference, $95.6\%$ of values are below 0.15, $99.7\%$ are below 0.2, approximately all values are below 0.3 and 0.4. The bivarite analysis also suggests a positive association ($\beta = 0.030$, $s.d = 0.049$, $p < 0.001$).

## G  DEFINITION OF THE ALTERNATIVE METRIC: WEIGHTED OVERLAP

In addition to the weighted cosine similarity, we report results with an alternative metric, **weighted overlap**, which measures the raw probability mass assigned to the feature-associated token set $T_f \subset V$ in Section H and I. This metric does not smooth over near misses via embedding similarity; instead, it directly captures how much of the model's next-token distribution lands on tokens in $T_f$.

**Definition.**  For a model output distribution $O \in \Delta^{|V|}$ and target token set $T_f$,

$$w(O, T_f) \;=\; \sum_{t \in T_f} O(t). \tag{3}$$

**Intervention effect.**  Let $O$ and $\tilde{O}$ denote the model's output distributions before and after intervention, respectively. The (absolute) change in weighted overlap is

$$\Delta w \;=\; w(\tilde{O}, T_f) \;-\; w(O, T_f). \tag{4}$$

**Relative change.**  When a scale-free summary is preferred, we also report the relative change:

$$\widehat{\Delta w} \;=\; \frac{w(\tilde{O}, T_f) - w(O, T_f)}{\max\{w(O, T_f), \epsilon\}}, \tag{5}$$

where $\epsilon > 0$ is a small constant to avoid division by zero.

# H  INTERVENTION TEST WITH FEATURE DIRECTION

## H.1  GENERALIZED FORMULATION OF A STEERING-VECTOR INTERVENTION

Let $x_{1:T} \in \{1, \cdots, V\}^T$ be the input sequence, $E \in \mathbb{R}^{V \times d}$ be the token-embedding matrix, and $\mathcal{G}_1$ to $\mathcal{G}_L$ be the blocks of a decoder-only Transformer. The unperturbed hidden states are

$$H_0 = E[x_{1:T}], \qquad H_\ell = \mathcal{G}_\ell \left(H_{\ell-1}\right) \qquad (\ell = 1, \ldots, L).$$

For any layer index $p$, we denote the vectorized activation as

$$A_p = \text{vec}\left(H_p\right) \in \mathbb{R}^{d \times T}.$$

With different strategies, we extract the steering direction $z_p \in \mathbb{R}^{d \times T}$. For injection at site $s$, we define the linear Jacobian:

$$\Phi_{p \to s} : \mathbb{R}^{d \times T} \to \mathbb{R}^{d_s}$$

obtained by composing linear portions between indices $p$ and $s$. The transported steering direction is

$$z_s = \begin{cases} \Phi_{p \to s} z_p, & \text{if } s > p \\ \Phi_{s \to p}^\dagger z_p, & \text{if } s < p \end{cases}$$

where $\dagger$ denotes the Moore–Penrose pseudo-inverse. When $s = p$, we set $z_s = z_p$. Eventually, we modify the activation at site $s$:

$$\widetilde{A}_s = A_s + \alpha z_s.$$

The network proceeds normally with this perturbation, yielding modified hidden states $\widetilde{H}_\ell$ and logits $\widetilde{y}_{1:T}$.

## H.2  EXPERIMENT DETAILS

To obtain the complete intervention data of various levels of interference features on the target features, we first filtered out all features in each layer that contained interference values at all levels, and for which the semantic similarity with the target feature is below the four selected thresholds. Subsequently, we select a subset of these features as the target features and identify the corresponding interference features across the various interference levels. Next, we search for other interference features with low interference values to the target features for experimentation. Specifically, other interference features are selected based on the interference values lying in intervals: $[0.0, 0.1]$, $[0.1, 0.2]$, $[0.2, 0.3]$ and $[0.3, 0.4]$. The scale parameter was tested within the range of $[-20, 20]$, and we avoid larger ranges to prevent severe disruption of the model.

Due to limitations on computational power, we sample clusters and features across various layers. In each SAE experiment with `Pythia-70M`, we sample 180 target features and collect approximately 2,700 interference features. In each SAE experiment with `GPT-2-Small`, we sample 480 target features and collect approximately $7,200$ interference features. For gradient experiments, we reduced 60% sampled features, but three times the number of gradient intervention vectors are extracted to keep the test set scale. As mentioned in the main text, for each feature, we focus on its top-activating token and use `DeepSeek-V3` to generate three prompt sentences for it. For each sentence, we test within the aforementioned scale range and record the result with the greatest improvement in the two metrics. This result means the best performance that the steering vector can achieve to induce the semantics of output toward the target feature without significantly disrupting the model. The final two metrics are averaged across all sentences for all features. To show the robustness of our experiments, the results of the alternative metric are presented in Figure 10.

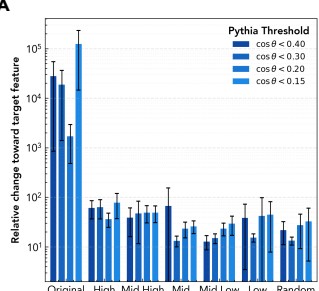 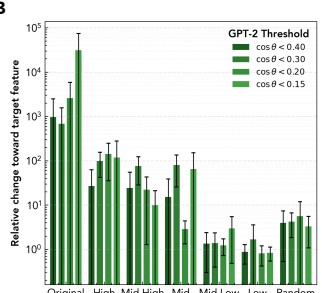 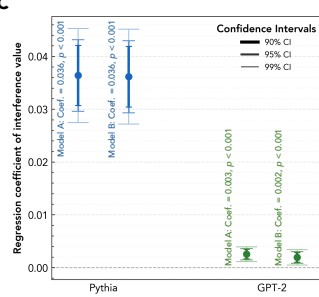

Figure 10: **Feature-level intervention effects measured by weighted overlap.** Subpanels follow the same conventions as Figure 3, but intervention effects are computed using relative change of weighted overlap $(\widehat{\Delta w})$. For (A–B), error bars indicate 95% confidence intervals; for (C), error bars denote 90%, 95%, and 99% confidence intervals.

## I INTERVENTION TEST WITH TOKEN'S GRADIENT

### I.1 TOKEN GRADIENT DIRECTION EXTRACTION

Given a tokenized input sequence $x = [x_0, \ldots, x_{T-1}]$, let $e_i = E[x_i]$ denote the embedding of token $x_i$, and $\mathbf{e} = [e_0, \ldots, e_{T-1}]$ the full input embedding sequence. Let $f_\ell : \mathbb{R}^{T \times d} \to \mathcal{A}_\ell^T$ denote the model's transformation up to layer $\ell$. The activation at position $i$ is:

$$a_{\ell,i} = f_\ell(\mathbf{e})[i] \in \mathcal{A}_\ell.$$

We define a scalar probe loss that selects this activation via a linear projection vector $v \in \mathbb{R}^d$:

$$\mathcal{L} = \langle a_{\ell,i}, v \rangle.$$

The gradient of this loss with respect to the input embedding $e_i$ is:

$$g_{\ell,i} := \frac{\partial \mathcal{L}}{\partial e_i} = \frac{\partial \langle a_{\ell,i}, v \rangle}{\partial e_i}.$$

We then normalize this vector to obtain a direction in embedding space:

$$\hat{g}_{\ell,i} := \frac{g_{\ell,i}}{\|g_{\ell,i}\|_2}.$$

We refer to $\hat{g}_{\ell,i}$ as the *token gradient direction*—the direction in input embedding space along which perturbations to token $x_i$ most increase its activation in $\mathcal{A}_\ell$ along $v$.

### I.2 EXPERIMENT DETAILS

Steering with the feature direction requires a pretrained sparse auto-encoder of the target model, which incorporates substantial computational costs and lacks scalability. To break this limitation, we need to explore a general approach. Observe that the SAE features are activated mainly by the top-activating token in its activation texts, while other tokens are just diluting its expression. Based on this observation, we can obtain a better steering vector by focusing on this particular token, and a sketch is as follows. We first feed the feature's activation text into the model, then compute the gradients of the top-activating token with respect to all neurons in the layer. The resulting gradients are combined to form a vector.

The SAE dataset from *Neuronpedia* contains approximately 50 activation text segments per active SAE feature, each strongly activating its corresponding feature. Due to computational limitations, we try to extract the gradient vectors from the first 3 activation texts of each feature. Also, we scale the vector within the same range $[0.5, 1, 1.5, 2, 3, 4, 5, 6, 7, 8, 9, 10, 12, 14, 17, 20]$. Each experiment on steering with token gradients combined with steering with feature directions can be done in one hour and a half for `Pythia-70M` and six hours for `GPT-2-Small`, running on a single thread of Intel i7-14700K. The results of the alternative evaluation metric are presented in Figure 11.

Table 3: **Examples of interventions using SAE features, token gradients, and prompt injections**

| Type | Model | Intervention | Target feature | Result |
|------|-------|--------------|----------------|--------|
| Feature | Pythia-70M | Steering feature vector: occurrences of specific surnames | Geographical locations | *"In the next week, we will go to"*
↑ Entered  ↓ Dropped
Berlin +0.025 our -0.029
London +0.012 some -0.012
To +0.010 an -0.010 |
| | GPT-2-Small | Steering feature vector: positive or negative event outcomes | Expressions of sadness | *"After hearing the bad news, she felt incredibly"*
↑ Entered  ↓ Dropped
grateful +0.051 bad -0.047
blessed +0.028 guilty -0.044
excited +0.028 uncomfortable -0.025 |
| Token | Pythia-70M | Steering token vector: legal terminology related to licenses and their implications | Elements related to political commentary and critique | *"In the election of this year, it is suggested to vote for"*
↑ Entered  ↓ Dropped
Donald +0.030 an -0.020
more +0.026 one -0.015
@ +0.015 President -0.013 |
| | GPT-2-Small | Steering token vector: key terms related to prices and transactions | References to location Tokyo | *"The organizing committee just announced that the upcoming finals will be held in"*
↑ Entered  ↓ Dropped
Tokyo +0.005 Toronto -0.007
Seoul +0.003 Seattle -0.005
Moscow +0.004 London -0.001 |
| | Llama-3.1-8b-Instruct | Steering gradient vector: references to the world and its various aspects | References to 'Switzerland' | *"I would like to recommend you to spend holidays in"*
↑ Entered  ↓ Dropped
Switzerland +0.16 Italy -0.034
Germany +0.089 Greece -0.016
Canada +0.015 Bulgaria -0.012 |
| Prompt | Pythia-70M | Injection of the tokens "Court" and "Dat", both before and within the text | References to locations | *"In the upcoming holiday, we will go to"*
↑ Entered  ↓ Dropped
Japan +0.021 some -0.014
Europe +0.015 an -0.006
Tokyo +0.012 see +0.003 |
| | GPT-2-Small | Prepending the injection text "(team writers writers)" | Terms related to names or surnames | *"After years of hard work, the award finally went to"*
↑ Entered  ↓ Dropped
Steve +0.005 China -0.006
John +0.002 waste -0.005
one +0.003 Donald +0.003 |
| | Llama-3.1-8b-Instruct | Prepending the injection text "(placement from placement)" | References to locations | *"In the next weekend we will go to"*
↑ Entered  ↓ Dropped
Paris +0.011 another -0.013
** +0.010 H -0.003
- +0.006 K -0.003 |

*Note*: ↑ Entered means that corresponding tokens entered the top-10; ↓ Dropped means that corresponding tokens dropped from the top-10. Gray-shaded rows indicate black-box interventions.

## J  EXPERIMENTS WITH DEEPSEEK-V3 FEATURE DESCRIPTIONS

To evaluate the sensitivity of our findings to the selection of LLMs for auto-interpretation, we employ an alternative model, DeepSeek-V3, to interpret the features of a sparse auto-encoder. The

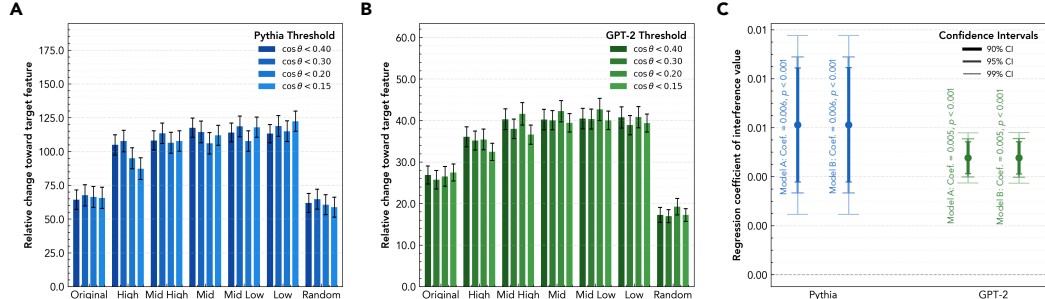

Figure 11: **Token-level intervention effects measured by weighted overlap.** Subpanels follow the same conventions as Figure 4, but intervention effects are computed using relative change of weighted overlap ($\widehat{\Delta w}$). For (A–B), error bars indicate 95% confidence intervals; for (C), error bars denote 90%, 95%, and 99% confidence intervals.

explanation text is subsequently processed by the `text-embedding-3-large` model to obtain embeddings. These embeddings are then utilized as semantic vectors to assess the similarity between features. For this illustrative sampling, features from the `Pythia-70M` att-5 layer are selected. The interpretation of features by large models is generated by feeding the model the top-activating texts of the feature and denoting the high activating tokens in it. Specifically, the prompts we write for `DeepSeek-V3` are as listed below.

> **System:** We're studying neurons in a neural network. Each neuron activates on some particular word or concept in a short document. The activating words in each document are enclosed by « and ». Look at the parts of the document the neuron activates for and summarize in a single sentence what the neuron is activating on. Try to be general in your explanations. Don't just repeat activation words. Also, you can summarize multiple points if the text content is not highly consistent. Pay attention to things like the capitalization and punctuation of the activating words or concepts, if that seems relevant. Keep the explanation as short and simple as possible, limited to 32 words or less. Omit punctuation and formatting.
> **User:** The activating documents are given below:
> 1.activation_text_1
> 2.activation_text_2
> ...
> 5.activation_text_5

In Figure 12, we compare semantic relatedness among SAE features using embeddings of explanations generated by `DeepSeek-V3` and `GPT-4o-mini`. The left density plot shows that pairwise similarities from the two models are tightly aligned, while the right heatmaps further illustrate that both models induce comparable feature–feature semantic structure.

Figure 13 compares SAE feature-direction and gradient-based interventions under auto-interpretations from `GPT-4o-mini` and `DeepSeek-V3`. Semantic dissimilarity threshold is set to different scales for intervention feature selection. Although this check is limited to a single layer (and thus exhibits greater variance), we consistently observe that high-interference, low-semantic-similarity features steer the target far more strongly than a random baseline. This pattern holds under both interpreters, indicating robustness to the choice of auto-interpretation model.

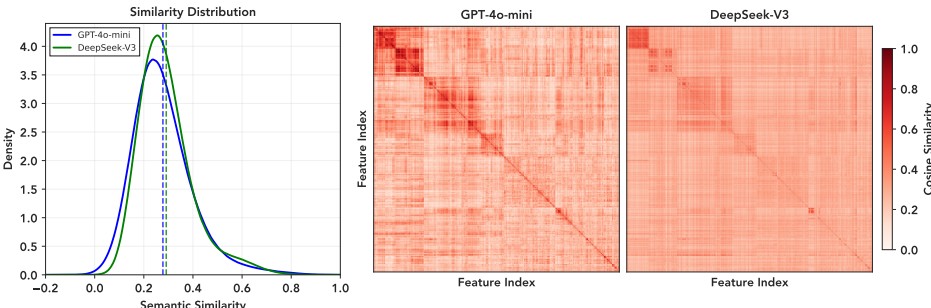

Figure 12: **Semantic relatedness between features from the view of `DeepSeek-V3` and `GPT-4o-mini`.**

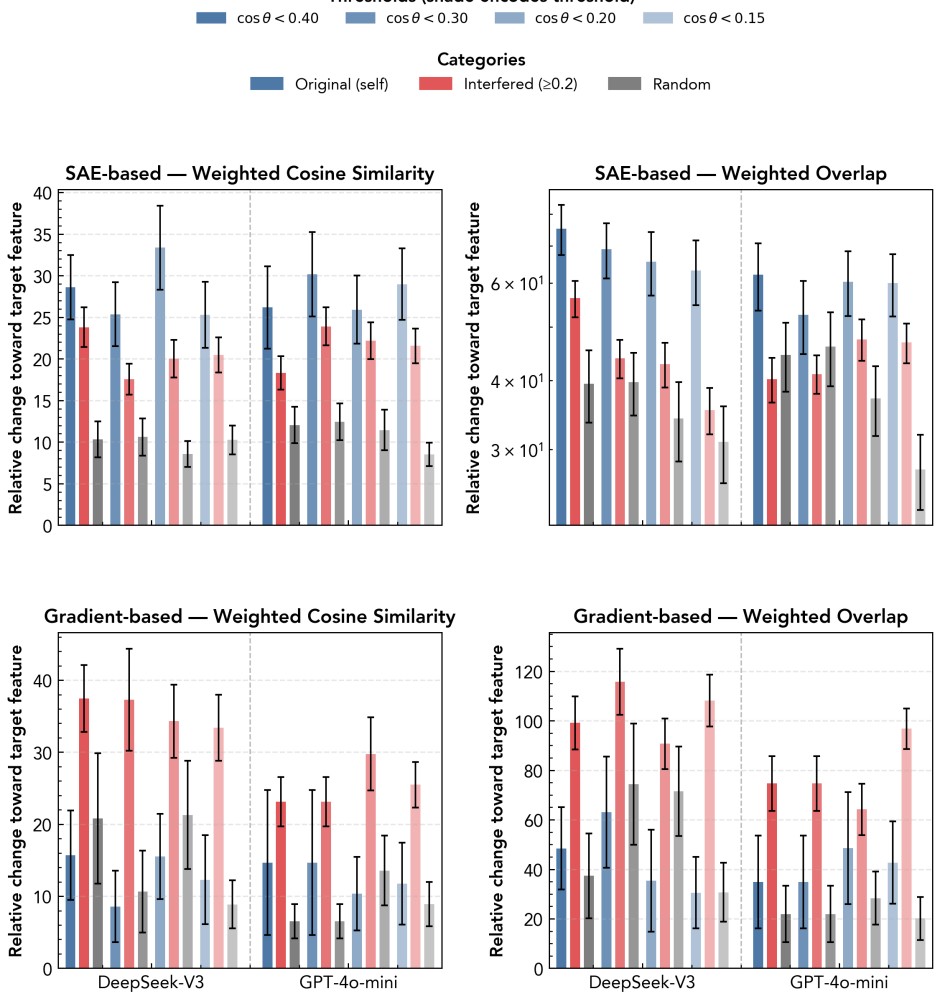

Figure 13: **Intervention effects under `GPT-4o-mini` vs. `DeepSeek-V3` auto-interpretations of SAE features.** Top row shows SAE-direction interventions; bottom row shows gradient-based interventions. Columns are the evaluation metrics: left, $\Delta c$; right, $\widehat{\Delta w}$. Error bars denote $95\%$ confidence intervals.

## K    BLACK-BOX INTERVENTIONS ON LARGER MODELS

We hypothesize that the polysemantic structures learned by large language models may exhibit some degree of generalizability. To explore this further, the interference study is extended to larger models without pretrained sparse auto-encoders.

### K.1    STEERING WITH TOKEN GRADIENT VECTOR

The scalable intervention on `Llama-3.1-8B-Instruct` is conducted by first selecting target type tokens as mentioned above and identifying target features in `Pythia-70M` and `GPT-2-Small` for which these tokens are the top-activating ones. The two interference feature sets in `Pythia-70M` and `GPT-2-Small` with respect to the target features are then identified. Next, we collect the top-activating tokens in two models respectively, and compute the union. The figure14 shows a sketch of the interference tokens extracted from two models.

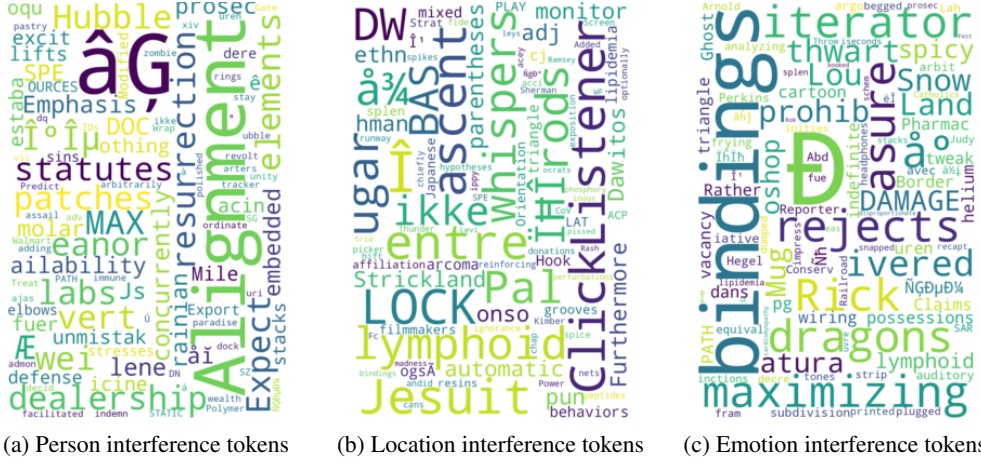

(a) Person interference tokens     (b) Location interference tokens     (c) Emotion interference tokens

Figure 14: **Three types of interference tokens**

After getting the tokens, we proceed to collect the activation texts, which may activate interference features of the target in black-box models. Due to computational constraints, we only compute gradients from and perform operations on the first half of residual layers in `Llama-3.1-8B-Instruct`. The intervention experiments are done for three target token types, each containing about 100 sentences. More intervention examples are listed below. It takes about 20 minutes for a single RTX4090 GPU to find a highly effective gradient vector for steering.

Table 4: **Examples of interventions on Llama-3.1-8B-Instruct Using Token Gradient Vector**

| Type | Intervention feature | Result |
|------|---------------------|--------|
| location | terms related to data and its presentation | *"After months of planning, our road trip finally reached"* 
 ↑ Entered     ↓ Dropped 
 New +0.017   an -0.006 
 Seattle +0.015   it -0.007 
 San +0.011   our -0.002 |
| | the verb "be" in various forms and contexts | *"She always dreamed of owning a small cafe in"* 
 ↑ Entered     ↓ Dropped 
 Vienna +0.080   France -0.008 
 Munich +0.071   town -0.007 
 Berlin +0.038   Italy -0.006 |

*Continued on next page*

Table 4 – **Examples of interventions Using Token Gradient Vector (Continued)**

| Type | Intervention feature | Result |
|------|----------------------|--------|
| | proper nouns, names, and references to specific roles or positions | *"This novel's opening scene takes place aboard a train to"* 
 ↑ Entered    ↓ Dropped 
 Beijing +0.023  New -0.026 
 Tokyo +0.020  Venice -0.009 
 Shanghai +0.018  Istanbul -0.010 |
| | quantitative data points related to statistics and performance metrics | *"The rebels established their hidden base deep within"* 
 ↑ Entered    ↓ Dropped 
 Afghanistan +0.018  an -0.009 
 Germany +0.015  their -0.003 
 Eastern +0.014  one -0.002 |
| | references to specific labeled items or categories | *"His last known coordinates placed him somewhere near"* 
 ↑ Entered    ↓ Dropped 
 Paris +0.012  an -0.003 
 New +0.010  their -0.003 
 Moscow +0.009  Lake -0.002 |
| person | quantitative data points related to statistics and performance metrics | *"Nobody expected the mysterious package to be from"* 
 ↑ Entered    ↓ Dropped 
 Paul +0.261  the -0.104 
 Emmanuel +0.026  a -0.078 
 Matthew +0.021  Lake -0.51 |
| | references to academic institutions or concepts | *"The voice on the recording definitely belongs to"* 
 ↑ Entered    ↓ Dropped 
 Robert +0.015  a -0.101 
 Patrick +0.012  the -0.088 
 David +0.009  me -0.033 |
| | phrases related to pre-approval processes and conditional statements | *"The fingerprints found at the scene match those of"* 
 ↑ Entered    ↓ Dropped 
 Michael +0.003  your -0.017 
 Richard +0.003  one -0.011 
 Smith +0.004  both -0.008 |
| | keywords related to file management and programming constructs | *"This traditional folk song was popularized by"* 
 ↑ Entered    ↓ Dropped 
 Bruce +0.010  Pete -0.086 
 Walter +0.010  American -0.032 
 Paul +0.007  Woody -0.021 |
| | terms related to multimedia and video production | *"The confidential information was leaked by former employee"* 
 ↑ Entered    ↓ Dropped 
 Mike +0.017  and -0.024 
 Tom +0.012  who -0.024 
 Bill +0.011  to -0.010 |
| emotion | instances of the verb "is." | *"After trying the new recipe, my brother absolutely"* 
 ↑ Entered    ↓ Dropped 
 love +0.121  fell -0.042 
 hate +0.095  LO -0.037 
 dislike +0.015  ad -0.031 |

Table 4 – **Examples of interventions Using Token Gradient Vector (Continued)**

| Type | Intervention feature | Result |
|---|---|---|
| | references to legal documents and real estate transactions | *"Science proves that most infants naturally"* 
 ↑ Entered     ↓ Dropped 
 Like +0.048   develop -0.099 
 like +0.020   prefer -0.051 
 love -0.001   learn -0.034 |
| | phrases indicating topics of discussion or content focus | *"His body language suggests he secretly"* 
 ↑ Entered     ↓ Dropped 
 love +0.457   wants -0.158 
 loved +0.012   enjoys -0.069 
 hate +0.015   hopes -0.062 |
| | phrases indicating relationships and affiliations in contexts such as surveillance, borders, and regulations | *"This fabric texture makes allergy sufferers"* 
 ↑ Entered     ↓ Dropped 
 love +0.126   miserable -0.091 
 like +0.020   feel -0.074 
 hate +0.027   and -0.043 |
| | statements that conclude or summarize concepts | *"After the concert critics began to"* 
 ↑ Entered     ↓ Dropped 
 hate +0.017   question -0.070 
 love +0.016   praise -0.064 
 enjoy +0.010   dissect -0.054 |

*Note*: ↑ Entered means that corresponding tokens entered the top-10; ↓ Dropped means that corresponding tokens dropped from the top-10. Gray-shaded rows indicate black-box interventions.

## K.2 PROMPT INJECTION

Beyond the two intervention methods discussed above, which directly manipulate the model's latent, we further examine whether models remain susceptible to polysemantic interference in more realistic inference-time scenarios. In such cases, interference may arise from the input prompt, which can indirectly influence the model's latent activations.

To begin with, following the same methodology, we first conduct an overall experiment in which we randomly sample target features across all sub-modules and layers along with their interference features at all levels. And for each interference feature, we select text snippets from its activation texts that include high activation tokens, and prepend them to the input prompt incorporated in two "**". To do a fine-tune on the snippet, we may add other zero-activation text besides it to maintain its grammatical or semantic structure, or just truncate its low-activation parts to see whether the interference effect is enhanced. We still make sure the snippets don't include tokens that are semantically related to the target feature. In this way, the interference potential of the interference feature to the target is roughly estimated, and we simply record the best case. Similarly, we use weighted cosine similarity and weighted overlap to quantify the semantic shift in model outputs toward the target feature. Note that comparing with the SAE and gradient-based intervention, we are using interference level 0.0-0.1 as random baseline instead of using snippets consisting of random tokens. This is because even random sampling of tokens may activate a feature such as "random tokens" or "spelling error", which has unknown interference level with the target feature, while the interference level of random baseline is explicit in the previous two experiments. Figure 15 shows the statistical results measured by alternative metric weighted overlap.

Case studies of prompt injection in our experiments are done on the token types listed in Appendix E. Based on the classification of tokens, we also generate test sentences for each type, requiring that these sentences grammatically lead to tokens of the target type. The prompt is as follows.

**System:** You are a creative writing assistant specialized in generating incomplete sentences. Your task is to create incomplete sentences where

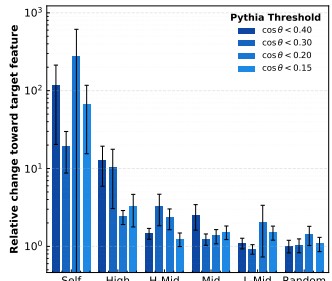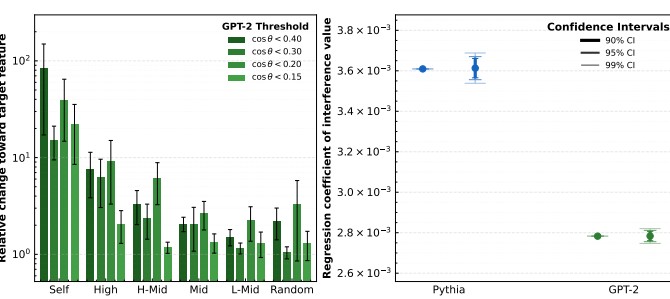

Figure 15: **Prompt injection effects measured by weighted overlap.** Relative change in weighted overlap toward the target. Bars show the mean relative change compared to baseline across interference levels, with lighter shades indicating stricter feature-meaning relevancy cutoff thresholds. The x-axis denotes the interference scale between the source features of injection snippets and the target feature. Error bars indicate 95% confidence intervals.

```
the next logical token would very likely be from the category category.

Requirements:
1.  Generate approximately 1000 incomplete sentences
2.  Each sentence should be approximately 20 tokens long when complete
3.  The sentences should end at a natural point where the next word would
very likely be a category word
4.  Use diverse contexts, scenarios, and grammatical structures
5.  Make the sentences engaging and varied
6.  Ensure the incomplete sentences create strong expectation for
category words

Examples for category:
- For 'animal':  "In the dense jungle, we could hear the roar of a wild"
- For 'color':  "The sunset painted the sky a beautiful shade of"
- For 'emotion':  "When she heard the news, her face showed pure"
- For 'location':  "Our vacation destination this summer will be"
- For 'number':  "The recipe calls for exactly"
- For 'person':  "The award ceremony will be hosted by a famous"
- For 'science':  "The experiment required careful measurement of"
- For 'time':  "The meeting is scheduled for next"

Return your response as a Python list of strings, with exactly 1000
sentences.  Format it properly as valid Python code that can be executed.
Start your response with:  sentences = [
End your response with:  ]

Do not include any explanatory text before or after the list.
```

`User:` `Generate approximately 1000 incomplete sentences for the category category.`

```
Available type tokens (separated by |, each token is enclosed in quotes):
type_token_1, type_token_2, ...

IMPORTANT NOTES:
- Each token above is a separate vocabulary item from language models
- Some tokens may have leading/trailing spaces (like " dog" or "cat ")
- These are the exact token strings that should be likely to appear as the
next token after your incomplete sentences
- Consider the token boundaries when creating sentences
```

```
Please generate diverse, engaging incomplete sentences where the next
word would very likely be from the category category tokens shown above.
Make sure to use various contexts and grammatical structures.

Return as a Python list:  sentences = [...]
```

The model returns approximately 400 to 800 sentences for each type. The dataset of token type denotation, along with the example sentences, has been made available in the GitHub repository.

After annotating the tokens in the model's vocabulary with type labels, we partition the token sets based on the intervention information provided by the sparse auto-encoder. First, we filter for features where the highly activated tokens contain the target token type, simply tagging these as target features. Specifically, for each activation text, we identify the token with the highest activation value (max_act) and set a ratio (here, 0.8). Tokens with activation values exceeding ratio * max_act are considered highly activating the interference feature.

We then identify all interference features whose interference value with the target feature exceeds 0.2 and whose semantic similarity is below 0.3. Based on this set of interference features, we further partition them into high-interference and medium-interference sets. Features with interference values above a high_threshold (for a given target feature) are classified as high-interference features, while those with interference values between 0.2 and high_threshold are classified as medium-interference features.

Subsequently, we collect the highly-activating tokens from features in each set to form the respective token sets. All remaining tokens that were not collected constitute the random token set. It should be noted that the high-interference and medium-interference token sets exhibit significant overlap. To address this, we deduplicate the two sets to obtain disjoint token sets. For example, in the case of the `Pythia-70M` model, we annotate 1,938 tokens as belonging to the "location" type. Using the method described above and setting 0.5 as the high-interference threshold, the resulting high-interference token set contains 10,185 tokens, while the medium-interference set consists of 34,946 tokens. And 13,323 tokens remain in the random set. There is an overlap of 9,535 tokens between the high- and medium-interference sets. After deduplication, the high-interference set retains 650 tokens, and the medium-interference set retains 25,404 tokens.

For experiments on `Pythia-70M` and `GPT-2-Small`, we use the token sets generated by each respective model. For experiments on `Llama-3.1-8B/70B-Instuct` and `Gemma-2-9B-Instruct` models, we adopt the union of the token sets from the corresponding interference levels of the two small models. We examine a total of eight categories of token types: location, person, emotion, color, animal, science, number, and time. Apart from the three types mentioned in the main text that demonstrate strong generalizability, the test results for the remaining types are presented below.

### K.2.1 POTENTIAL SHARED POLYSEMANTIC INTERFERENCE STRUCTURES

Based on preliminary token insertion tests, we first identify the most effective test cases by selecting those where high-interference token sets produce the most pronounced enhancement relative to medium- low-interference, and random sets. We then analyze these selected cases to trace back to the underlying interference features, which include them as highly-activating tokens. Cross-model comparison based on semantics of these interference features, which are quantified as stated in Section 2.1, further reveals that some of them occur in multiple models. A list of these recurring polysemantic feature pairs is provided below, along with their potential locations in each model. Note that for smaller models, positional hints are derived directly from sparse autoencoders; for larger models, locations are estimated by comparing activation patterns with a succinct, semantically neutral prompts (e.g., "In the weekend" for location-type features).

Table 5: **Token Types without Strong Generalizability**

| Target | Model | Original | High-interference | Low-interference | Random |
|---|---|---|---|---|---|
| Person | Pythia-70M | 60.28%*** | 29.02% | 28.58% | 32.31% |
| | GPT-2-Small | 54.29%*** | 27.47%*** | 26.97%** | 25.16% |
| | Llama-3.1-8B-Instruct | 38.85%*** | 19.20%*** | 19.08%*** | 16.86% |
| | Gemma-2-9B-Instruct | 43.27%*** | 25.36% | 26.10% | 25.04% |
| | Llama-3.1-70B-Instruct | 46.27%*** | 21.90%*** | 21.13%** | 19.61% |
| Animal | Pythia-70M | 96.96%*** | 28.49% | 30.75% | 29.83% |
| | GPT-2-Small | 86.11%*** | 21.64% | 18.87% | 25.03% |
| | Llama-3.1-8B-Instruct | 67.07%*** | 42.76%* | 40.56% | 40.89% |
| | Gemma-2-9B-Instruct | 26.09%*** | 41.02%*** | 38.50% | 38.47% |
| | Llama-3.1-70B-Instruct | 49.32%*** | 32.80%* | 30.79% | 31.43% |
| Emotion | Pythia-70M | 61.74%*** | 26.13%*** | 21.11% | 20.68% |
| | GPT-2-Small | 58.84%*** | 36.91%*** | 35.22%* | 33.85% |
| | Llama-3.1-8B-Instruct | 60.16%*** | 27.94% | 27.45% | 29.37% |
| | Gemma-2-9B-Instruct | 56.55%*** | 13.85% | 13.63% | 13.50% |
| | Llama-3.1-70B-Instruct | 51.51%*** | 44.90%*** | 40.63% | 40.40% |
| Color | Pythia-70M | 97.31%*** | 17.89% | 21.49% | 21.80% |
| | GPT-2-Small | 85.47%*** | 35.57% | 33.13% | 36.93% |
| | Llama-3.1-8B-Instruct | 76.76%*** | 22.01% | 20.67% | 20.97% |
| | Gemma-2-9B-Instruct | 22.57%*** | 13.58% | 16.10% | 15.67% |
| | Llama-3.1-70B-Instruct | 76.85%*** | 19.08% | 17.88% | 18.44% |
| Time | Pythia-70M | 69.92%*** | 45.33%*** | 45.54%*** | 41.33% |
| | GPT-2-Small | 58.48%*** | 31.55%*** | 27.72%*** | 21.42% |
| | Llama-3.1-8B-Instruct | 51.41%*** | 25.58% | 26.29% | 25.95% |
| | Gemma-2-9B-Instruct | 55.15%*** | 25.10%* | 25.94%*** | 23.62% |
| | Llama-3.1-70B-Instruct | 78.28%*** | 36.87% | 36.68% | 37.71% |

*Note*: Cell values show the success rate of elevating target-type tokens into the top 30 predictions. Gray-shaded rows indicate black-box interventions. Testing uses a shared token set from the two small models. ***, **, and * denote t-test significance at $p < 0.001$, $p < 0.01$, and $p < 0.05$, respectively, vs. random baseline. High- and low-interference tokens lie in $[0.5, 1.0]$ and $[0.2, 0.5]$.

Table 6: **Shared Polysemantic Interference Structures**

| Interference Feature Pairs | Position in Models |
|---|---|
| references to specific **locations** and establishment & datatypes, definitions or other concepts in programming context | Pythia-70M(res-4), GPT-2-Small(res_mid-6), Llama-3.1-8B(res-10), Gemma-2-9B(res-32), Llama-3.1-70B(res-22) |
| numerical values and their patterns & phrases related to emotional states or expressions | Pythia-70M(mlp-1), Llama3.1-8B(mlp-23), Gemma-2-9B(mlp-34) |
| references to medical and cellular biology & patterns of punctuation and formatting | Pythia-70M(res-0), GPT-2-Small(res_post-2), Llama-3.1-8B(res-30), Gemma-2-9B(res-11) |

## L    COVERT INTERVENTION ON *Hellaswag* TESTSET

To further examine the impact of the interference vectors obtained in the aforementioned sections on the overall performance of the model, we conduct a rapid validation using the experimental results targeting the "location" type of interference in `Llama-3.1-8B-Instruct`. Given that the intensity of the interference vectors applied in the previous experiments is aimed at maximizing the disruptive effect—which likely caused substantial impairment to the model, we first reduce the scale to $0.25$ times its original value. Subsequently, we randomly select $500$ test samples from the *HellaSwag* validation set and evaluate each interference vector on them. Using accuracy as the evaluation metric, the experimental results demonstrate that, out of a total of $174$ interference vectors, $149$ decreased the accuracy, while $25$ increased it. The baseline accuracy is $77.2\%$, and the average reduction in accuracy is $2.77\%$. Some examples that keep model performance, i.e. reduce accuracy fewer than $1\%$, while still having substantial intervention effects are listed below.

Table 7: **Covert Intervention on Hellaswag Dataset**

| Intervention Feature | Top Predicted Tokens | | | |
|---|---|---|---|---|
| Description of precautions related to safety protection | *"The documentary crew disappeared while filming in remote areas of"* | | | |
| | Raw | | Intervention | |
| | the | 0.35 | the | 0.24 |
| | Papua | 0.027 | Papua | 0.04 |
| | Nepal | 0.016 | Africa | 0.035 |
| Numerical data that may require ordering or sorting | *"She always dreamed of owning a small cafe in"* | | | |
| | Raw | | Intervention | |
| | the | 0.34 | the | 0.36 |
| | a | 0.27 | a | 0.28 |
| | her | 0.21 | Paris | 0.13 |

# M  TRANSFERABLE POLYSEMANTIC STRUCTURE EXPLORATION

## M.1  BRAINSTORMING HIDDEN ASSOCIATIONS

In this section, we analyze shared interference patterns in `GPT-2-Small` and `Pythia-70M`, which we previously demonstrated transfer to larger models under intervention. First, we use `DeepSeek-V3` and `GPT-5-mini` to annotate every selected feature pair with the following prompt instruction.

**System:** You are an expert in analyzing neural network feature semantics. Given two SAE (Sparse Autoencoder) features, their explanations, and their top activation texts, determine if they are semantically related. For each feature, you will see:
- An explanation describing what the feature captures
- 5 text segments where the feature activates most strongly. High-activation tokens are marked with «<token»> or «<multiple tokens»>

Analyze the explanations and the marked tokens/activation patterns to determine:
1.  Are these two features semantically related? Consider any form of semantic relationship – including direct overlaps (capturing similar concepts, linguistic patterns, or contextual meanings) as well as higher-order associations (e.g., semantic priming, thematic relatedness, or complementary roles).
2.  If related, provide a concise description (max 50 tokens) of their relationship.
Return your analysis in this exact JSON format:
"isRelated": true/false, "description": "brief description" or null
Examples:
- Features activating on different tenses of verbs: "isRelated": true, "description": "Both capture verbal expressions, one for past tense, other for present tense"
- Features for numbers vs. animals: "isRelated": false, "description": null
- Features for positive vs. negative emotions: "isRelated": true, "description": "Both capture emotional expressions with opposite valence"
- Features for "doctor" vs. "hospital": "isRelated": true, "description": "Conceptually linked via medical domain (profession vs. location)"
- Features for "boat" vs. "sand": "isRelated": true, "description": "Loosely associated through beach/marine context (object vs. terrain)"

**User:** Feature A: feature_a
Explanation:  feature_a's explanation
Top activations:  texts that activates feature_a the most
Feature B: feature_b Explanation:  feature_b's explanation
Top activations:  texts that activates feature_b the most

To compare the behavior of the two automatic annotators (`DeepSeek-V3` and `GPT-5-mini`), we compute the share of feature pairs labeled as "related" and the simple percent agreement in two subsets. In the `Pythia-70M` subset, `DeepSeek-V3` labels 10.8% of pairs as related and `GPT-5-mini` 28.7%, with 80.3% percent agreement. In the `GPT-2-Small` subset, the corresponding proportions are 3.7% (`DeepSeek-V3`) and 18.4% (`GPT-5-mini`), with 83.8% agreement. Aggregating both subsets, `DeepSeek-V3` labels 9.5% of pairs as related and `GPT-5-mini` 26.8%, and the overall percent agreement was 80.9%. We define percent agreement as the proportion of pairs for which both annotators assign the same label. Figure 15 A-B shows the agreement matrices for the two annotators. Interestingly, the overall consistency between the two annotators is low (`Pythia-70M`: Cohen's $k = 0.408$; `GPT-2-Small`: Cohen's $k = 0.221$). Compared with `DeepSeek-V3`, `GPT-5-mini` labels significantly more feature pairs as related, suggesting greater sensitivity to latent semantic

associations. Nevertheless, more than 70% of pairs are still judged completely unrelated in the annotators' combined judgment (i.e., both say unrelated).

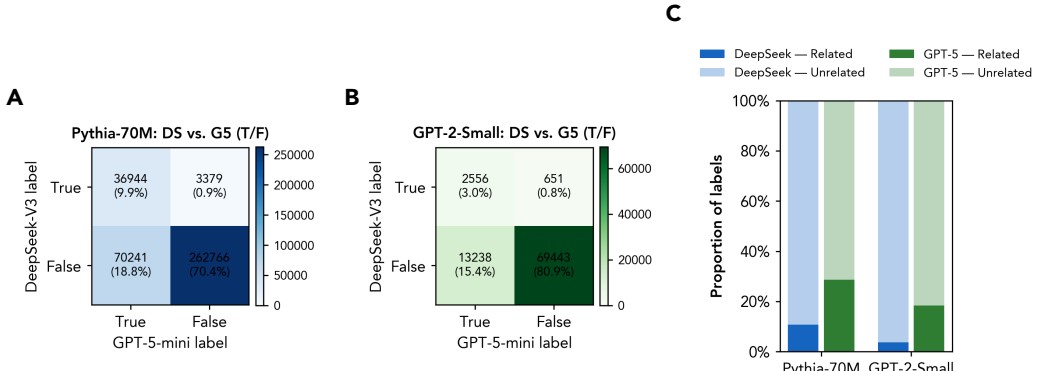

Figure 16: **`GPT-5-mini` and `DeepSeek-V3`'s annotation reports for shared interfered feature pairs.** DS is the abbreviation of `DeepSeek-V3` and G5 is the abbreviation of `GPT-5-mini`.

In Table 8, we further report eight qualitative examples where the two annotators flag latent associations and contrasting cases where they do not.

Table 8: **Eight examples of annotated feature pairs.**

| Category | Feature pair | Interpretation |
|---|---|---|
| Frame | **Feature A:** Phrases and sentences that highlight systemic issues related to incarceration and its effects on individuals and families.
**Feature B:** Quantitative information related to statistics and predictions. | Incarceration content often appears alongside quantitative/statistical references. (from `GPT-5-mini`) |
| Orthography | **Feature A:** Occurrences of the word "acher."
**Feature B:** References to specific researchers or authors in studies related to minimum wage. | Both capture researcher/author name substrings (e.g., -acher / Wascher). (from `GPT-5-mini`) |
| Axiology | **Feature A:** The term "dear" in emotional contexts related to relationships and feelings of affection.
**Feature B:** Concepts related to the notion of the sacred. | Both capture emotionally significant concepts (affection vs. sacredness) with deep personal or spiritual value. (from `DeepSeek-V3`) |
| Homography | **Feature A:** References to playing cards and card-related concepts.
**Feature B:** References to the authors or studies related to economic analysis. | Surface lexical overlap: "card" as playing card vs "Card" (author name). Same token, different senses. (from `GPT-5-mini`) |
| No-relation | **Feature A:** References to "New Guinea."
**Feature B:** File formats and file compression terminology. | None |
| No-relation | **Feature A:** Instances of the abbreviation "ob" or related terms indicating observational data or annotations.
**Feature B:** A specific term related to a well-known ride-sharing company. | None |
| No-relation | **Feature A:** Terms related to turbidity and its measurement.
**Feature B:** References to specific music artists or groups. | None |
| No-relation* | **Feature A:** Occurrences of the name "Beethoven" and related variations.
**Feature B:** Words related to expressions of frustration or annoyance. | None |

## M.2 FEATURE-PAIR COMPARISON TEST

Let $s_{\mathcal{M}}(f_i, f_j)$ denote the semantic similarity (in $\mathcal{M}$) between features $f_i$ and $f_j$, and let $I(f_i, f_j)$ denote their interference value.

From the set of shared interfering feature pairs in `GPT-2-Small` and `Pythia-70M` satisfying

$$s_{\mathcal{M}}(f_i, f_j) < 0.4 \quad \text{and} \quad I(f_i, f_j) > 0.4,$$

we uniformly sample pairs for testing. For each sampled pair $(f_1, f_2)$, we then search for a matched comparison pair $(f_3, f_4)$ from the same layer such that

$$\left| s_{\mathcal{M}}(f_3, f_4) - s_{\mathcal{M}}(f_1, f_2) \right| \leq \varepsilon \quad \text{and} \quad I(f_3, f_4) < 0.1,$$

i.e., semantic similarity is tightly matched while interference is substantially lower. For `Pythia-70M`, $\varepsilon = 0.01$; for `GPT-2-Small`, due to limited matching, $\varepsilon = 0.05$. Eventually, we collect $3,800$ feature-pair combination for this comparative analysis.

We then present the two pairs $(f_1, f_2)$ and $(f_3, f_4)$ (order randomized) to `GPT-5-mini`, asking the model to choose the more related pair using the prompt below.

> **System:** You are an expert in interpretability of LLM SAE features.  You
> will receive two feature pairs (pair_one, pair_two).  For each pair (f,
> g), use their explanations and Top-5 highest-activation texts (with
> high-activation tokens marked with «<token>» or «<multiple tokens>») to
> decide which pair shows higher latent relatedness.
>
> Definition:
> "Latent relatedness" means the two features in a pair likely reflect the
> same underlying concept or are functionally coupled.  Consider direct
> overlaps (concepts, patterns, meanings) and higher-order associations
> (thematic/complementary roles).
>
> Decision criteria:
> 1) Explanations:  semantic consistency, paraphrase/synonymy, or
> complementary roles within the same domain; penalize opposite/disjoint
> concepts.
> 2) Top-5 texts:  overlap in topics/contexts; alignment of «<markers»>
> pointing to the same phrase, slot, entity, or role; penalize disjoint or
> contradictory evidence.
>
> Uncertainty handling:
> If evidence from (1)-(2) is insufficient or conflicting, return
> "uncertain".
>
> Output:
> Return ONLY a JSON object with exactly these keys:
> "more_related_pair":  "pair_one" or "pair_two" or "uncertain"
> "reason":  brief English rationale ($\leq$ 80 words), citing 1-2 decisive cues
> (e.g., explanation alignment, shared topics, token markers).
> Must be valid JSON (double quotes).  No extra text.

> **User:** PAIR ONE
> Feature 1
> Explanation:  explanation of feature one
> Texts:  five activation texts of feature one
>
> Feature 2
> Explanation:  explanation of feature two
> Texts:  five activation texts of feature two
>
> PAIR TWO

```
Feature 1
Explanation:  explanation of feature one
Texts:  five activation texts of feature one

Feature 2
Explanation:  explanation of feature two
Texts:  five activation texts of feature two
```

## N  POLYSEMANTIC NEURON MANIPULATION

During the examination of SAE features and their connections with neurons, many features exhibit semantically similar activation texts. To avoid repetitive analysis on similar activation texts, we first perform feature clustering based on the semantics of activation texts, and then check neuron connection at the cluster level. Given that the sparse auto-encoder from *Neuronpedia* is trained with a sparsity setting of 3, the analysis focuses on the top three neurons with the highest alignment values per cluster. A threshold of 0.2 is applied to filter out weak connections. Figure 17 shows the distribution of polysemantic neurons identified in each layer. We can see that polysemantic neurons with strong connections with aggregated features only take up fewer than 5% in each layer.

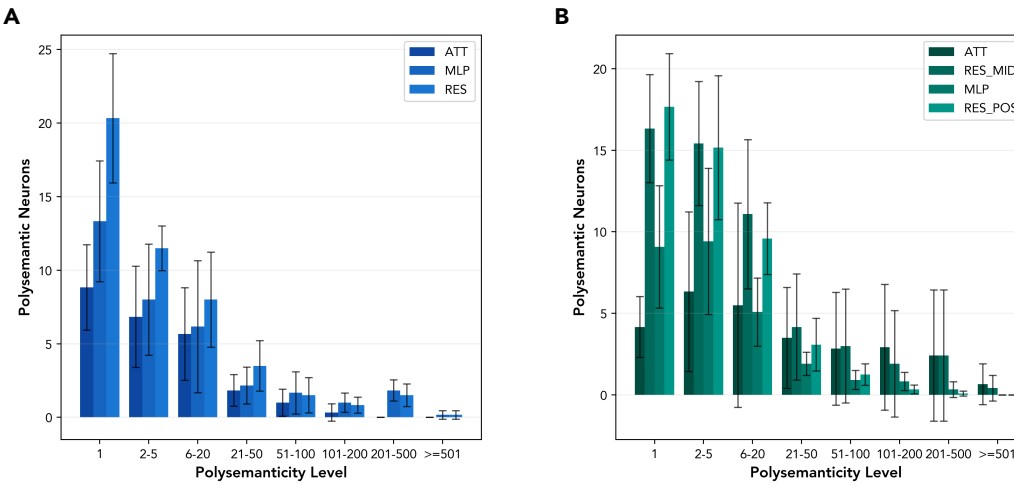

Figure 17: **Distribution of polysemantic neurons in each model.** A is the result of `Pythia-70M`, and B is the result of `GPT-2-Small`. Error bars represent 95% confidence intervals.

For strongly connected polysemantic neurons, we do further investigations on how suppressing or boosting their activation influences the semantic shift in the model's output to their aligned features. Neurons' activation is multiplied with a scale value in the range $[0, 20]$. Note that scaling within $[0, 1]$ suppresses activation, while scaling within $[1, 20]$ amplifies it.

## O  DENOISING INTERVENTION EFFECT

We observe that the SAE intervention experiments on `GPT-2-Small` yield considerably weaker effects compared to the `Pythia-70M` model, although the overall trend—where high-interference intervals tend to produce stronger effects—is still maintained. Given that our experimental results represent an average over test cases from all layers and all sub-modules of the model, including both effective and ineffective instances, we hypothesize that the presence of ineffective cases in `GPT-2-Small` may account for this outcome.

Therefore, we apply a simple preprocessing step to the test cases and observe a marked improvement ($5\times$ in `Pythia-70M` and $100\times$ in `GPT-2-Small`) in the experimental results. Specifically, we first employ the direction of the target feature for an initial steering attempt, monitoring whether the two metrics improve after steering. We set thresholds, approximately between 0.05 and 0.1, to filter ineffective feature directions. Subsequently, we select interference vectors for the remaining potential directions and proceed with the experiments. The results are shown in Figure 18.

We find that with this simple preprocessing step, the performance gap between `GPT-2-Small` and `Pythia-70M` narrows to approximately tenfold, compared to the original hundredfold difference. To explore variations in interference effects across different models, we select an additional model, `Gemma-2-2B`, and conduct corresponding interference tests for a subsample. The specific results are presented below in Figure 19.

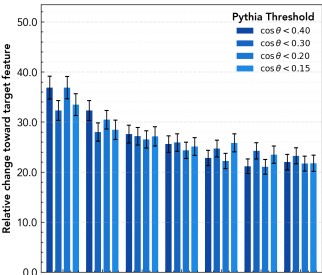 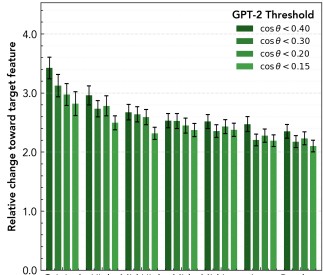 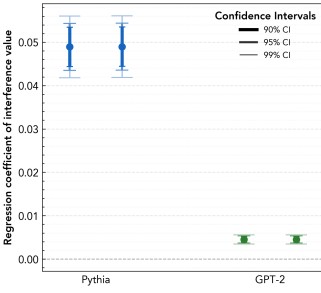

Figure 18: **Enhanced Effects of intervention based on the interference SAE feature direction.** (A–B) Relative change in weighted cosine similarity toward the target ($\Delta c$). Bars show the mean relative change compared to baseline across interference levels, with lighter shades indicating stricter feature-meaning relevancy cutoff thresholds. The x-axis denotes the interference scale between the target and intervention feature: *Original* corresponds to intervening with the target feature itself, and *Random* serves as a random feature intervention baseline. Error bars denote 95% confidence intervals. (C) Regression estimates of the effect of feature-pair interference value on intervention success. Two regression specifications are shown: Model A regresses weighted cosine similarity after intervention ($c(\tilde{O}, T_f)$) on interference value, with feature-meaning similarity, baseline weighted cosine similarity ($c(O, T_f)$), and layer-type controls; Model B regresses the change score ($\Delta c$) on interference value, with feature-meaning similarity and layer-type controls. Error bars denote 90%, 95%, and 99% confidence intervals.

Comparing the two metrics, the three models exhibit considerable differences under the weighted cosine similarity metrics. However, under the weighted overlap metric, the high-interference intervals in all three models demonstrate comparable effects, with an improvement of approximately two orders of magnitude (about $10^2$). We therefore conclude that interference in `GPT-2-Small` more precisely enhances the probability of the tokens most relevant to the target feature in the output.

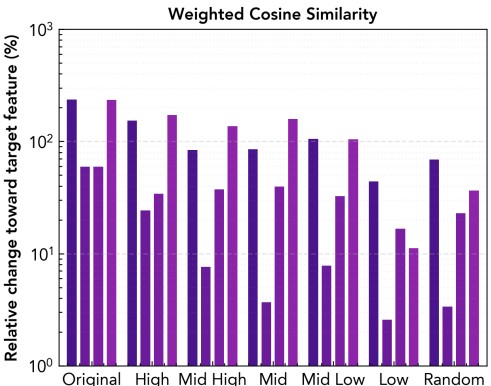 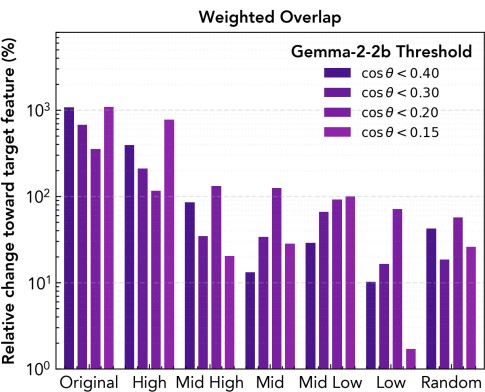

Figure 19: **Effects of SAE intervention on `Gemma-2-2B`.** Relative change in weighted cosine similarity and weighted overlap towards the target ($\Delta c$, $\Delta o$). Bars show the mean relative change compared to baseline across interference levels, with lighter shades indicating stricter feature-meaning relevancy cutoff thresholds. The x-axis denotes the interference scale between the target and intervention feature: *Original* corresponds to intervening with the target feature itself, and *Random* serves as a random feature intervention baseline.

## P    ETHICS STATEMENT, LIMITATIONS AND FUTURE WORKS

This paper follows the ICLR Code of Ethics. This study has three key methodological limitations. First, we rely on SAEs to disentangle polysemantic activations; although SAEs are the de-facto tool, their outputs fluctuate with dimensionality and hyper-parameters, yielding unstable features (Paulo & Belrose, 2025; Heap et al., 2025; Gao et al., 2024). Second, our interventions steer only one interference feature in one layer, while multi-feature, cross-layer manipulations could amplify and better obscure the effect (Ameisen et al., 2025). Third, we quantify vulnerability solely via shifts in immediate next-token probabilities on two small base models—because only they both expose raw logits and have pre-trained SAEs—then check coarse transfer on three larger instructed models; establishing how these interventions alter non-trivial downstream tasks in bigger models is the next stage of this project.

## Q    REPRODUCIBILITY STATEMENT

To balance reproducibility with responsible disclosure, we release complete code, evaluation scripts, and synthetic data in this Github repository, but deliberately omit the matrices that catalogue shared polysemantic directions between two small models. Publishing those mappings would make it easier to weaponize the very vulnerabilities we study, whereas the available artifacts still permit independent verification of all empirical claims.

