# OpenReview forum: "Signal in the Noise: Polysemantic Interference Transfers and Predicts Cross-Model Influence"
_ICLR.cc/2026/Conference — ICLR 2026 Poster_

### Official Review · Reviewer_dp5x · 2025-10-26

**Soundness:** 3
**Presentation:** 2
**Contribution:** 3
**Rating:** 6
**Confidence:** 4

**Summary:**

This work investigates polysemanticity in language models using SAEs, and whether this polysemanticity translates to practical vulnerabilities, where vulnerabilities are characterised based on shifts in next token predictions following interventions. They identify pairs of features which interfere, but are semantically unrelated, and show that black and white box interventions (prompt injection and steering) made on one feature interfere with the other. Interestingly, they find that these interference pairs, which appear to have no interpretable semantic link, replicate across models of different sizes, as shown with steering experiments. Regarding vulnerabilities, they show that altering the activations of more polysemantic neurons leads to greater shifts in outputs.

This is an interesting and well conducted investigation into a known but minimally explored implication of superposition. The writing and Figures are unclear in places (I have made some suggestions to improve this) and this does detract from the contributions. Furthermore, some of the relationships observed in the results are unclear and merit more explanation and investigation, as I have raised in the weaknesses and questions. However, overall this is an interesting piece of work with potentially valuable practical implications.

**Strengths:**

This paper addresses a potentially impactful but minimally studied issue: the impact of polysemanticity on LLM vulnerability.

Good and thorough analysis, both within the main text (e.g. running experiments with features sampled from different interference intervals) and in the additional results in the Appendices (e.g. the correlation results between interference and semantic similarity in Figure 8 are an interesting additional result). Multiple metrics are used to validate results, and results are reported for multiple models with confidence intervals.

The set of black and white box interventions used are well-designed and thorough. The gradient-based steering approach, in particular, is a clever way to run steering experiments on models where SAEs are not available.
The finding that polysemantic structures transfer across models, despite having no immediately interpretable meaning, is quite novel and opens up interesting directions for further investigation.

Where there is confusion around the results (e.g. the major gap between models in Figure 3), the authors are transparent and honest, which I appreciate.

**Weaknesses:**

The introduction is long and seems to try and incorporate a large amount of related work (while the main RW section is in the Appendices). This makes it difficult to follow the purpose of the paper. I’d recommend using the additional page to change this to a condensed introduction followed by a separate related work.

I found Figure 1 confusing, especially as it appears before the representation domains are defined. The text on the right is a useful summary, but it was not immediately clear to me what the top boxes were trying to show. Could you move these below the other graph (so the reader reads the text first) and/or add labels to these boxes? Please also increase the font sizes in Figures 2-4.

The SAE feature steering results (Figure 3) are extremely different between Pythia and GPT-2, and I don’t find the hypothesised cause - model depth - to be particularly convincing. It would be great to repeat this with a 3rd model to compare the results, or at least to discuss or investigate other possible reasons in more detail (e.g. it may be SAE quality, since the difference in effect size is much smaller with the gradient direction interventions).

This paper could also be improved by adding a realistic case study of how interference values can be used to extract examples of prompt injections which elicit dangerous responses. This would be particularly impressive if they transfer to closed weight models.

**Questions:**

What is your intuition behind the weighted cosine similarity metric (Equation 1) and how what this measures differs to the weighted overlap? Why select these measures?

Why do you use token frequency instead of these metrics in the prompt injection experiments?

Please define interference when you first refer to it in the introduction - its somewhat ambiguous otherwise. Similarly for the ‘feature irrelevance thresholds’ introduced in Section 3.1 - it would be much clearer if you define and link this to the semantic similarity measure in the main text (its also not clearly described in the Appendix).

Can you repeat the experiment in Figure 3 with a third model, to investigate the 100x difference in magnitude between the Pythia and GPT-2 effects?

The results on Figures 3 and 4 show a 100x difference between these methods for GPT-2 but the text in Section 3.2 says 10x - is this a typo or error on one of the graphs?

Can you move the note for Table 1 into the caption? The trends may also be clearer if you report the change in % relative to the random token rather than the absolute values. Also increase the shading contrast as this is barely visible!

Can you add interference and semantic similarity scores to Table 3?

Minor comment - the reference on line 143 is misformatted.

---

> ### Author Response · Authors · 2025-11-21
>
> Thank you for your thoughtful review, including your appreciation of our work and your insightful comments regarding some of the smaller effect sizes. In the revised manuscript, we have incorporated changes reflecting your suggestions, with all modifications highlighted in red; below, we respond to your main concerns and questions in detail.
>
> **W.1 The introduction is long and seems to try and incorporate a large amount of related work (while the main RW section is in the Appendices). This makes it difficult to follow the purpose of the paper. I’d recommend using the additional page to change this to a condensed introduction followed by a separate related work.**\
> Thank you for this suggestion and for flagging the structural issue. We agree that the current introduction is overly long and mixes core motivation with an extended discussion of related work, which can obscure the main purpose of the paper. In the final revised manuscript (by December), we will (i) shorten and refocus the introduction on the problem statement, key contributions, and high-level intuition, and (ii) move most of the literature discussion into a dedicated **Related Work** section that follows the introduction.
> \
> \
> **W.2 I found Figure 1 confusing, especially as it appears before the representation domains are defined. The text on the right is a useful summary, but it was not immediately clear to me what the top boxes were trying to show. Could you move these below the other graph (so the reader reads the text first) and/or add labels to these boxes? Please also increase the font sizes in Figures 2-4.**\
> Thank you for your suggestions on improving the readability of our work. We will organize the figures and descriptions well in our final revision.
> \
> \
> **W.3&Q.4 The SAE feature steering results (Figure 3) are extremely different between Pythia and GPT-2, and I don’t find the hypothesised cause - model depth - to be particularly convincing. It would be great to repeat this with a 3rd model to compare the results, or at least to discuss or investigate other possible reasons in more detail (e.g. it may be SAE quality, since the difference in effect size is much smaller with the gradient direction interventions).**\
> Thank you for this thoughtful question and for raising the concern about the discrepancy between Pythia-70M and GPT-2-Small in Figure 3. You are right that the SAE-steering effects on GPT-2-Small appear modest in the original draft. One important reason is that the reported numbers are averages across all layers and submodules, which mix together both effective and ineffective steering directions.
>
> In the revision, we introduce a simple filtering step to remove clearly ineffective steering directions. This substantially increases the measured effect sizes: under our weighted cosine similarity metric, the average effect increases by about 8x for Pythia-70M and about 100x for GPT-2-Small, reducing the gap between the two models from roughly 100x to about 10x. The details of this procedure and the updated results are reported in Appendix O. Importantly, even in the original setting, the weighted overlap metric (Fig. 9) already showed comparable effect sizes across the two models, indicating that both can reliably enhance the most related tokens of the target feature in their outputs.
>
> Motivated by your suggestion, we also ran an additional SAE-steering experiment on a third model, Gemma-2-2B, on a smaller evaluation set. Despite its greater depth relative to GPT-2-Small, Gemma-2-2B shows a larger effect under weighted cosine similarity than GPT-2-Small, but still a smaller effect than Pythia-70M (Appendix O). At the same time, its improvements under the weighted overlap metric are similar to those of the two smaller models. Taken together, these results suggest that model depth alone cannot fully explain the observed differences in cosine-based effect sizes. In the revision, we therefore soften our earlier depth-based interpretation and explicitly present the precise cause of these differences as an open question, while emphasizing that cross-model transfer of interference structure is robust across all three models under our overlap-based metric.

---

> > ### Author Response · Authors · 2025-11-21
> >
> > **Q.1 What is your intuition behind the weighted cosine similarity metric (Equation 1) and how what this measures differs to the weighted overlap? Why select these measures?**\
> > Thank you for asking about the intuition behind our evaluation metrics. We realize this was not explained clearly enough in the current draft.
> >
> > Weighted cosine similarity (Eq. 1). Intuitively, this metric is designed to be sensitive to fine-grained shifts in the overall output distribution. When we steer toward a feature (e.g., “sadness”), we do not only care about a single canonical token, but about a whole cloud of semantically related expressions (“depressed”, “blue”, “unhappy”, etc.), including minor variations such as casing or leading spaces. Weighted cosine similarity aggregates these changes into a single continuous similarity score, capturing smooth re-weighting of many related tokens in the full distribution.
> >
> > Weighted overlap. In contrast, the weighted overlap metric is intentionally more “coarse” and focuses on large, discrete changes in the probability mass assigned to the most relevant tokens for a feature. It asks, roughly: after intervention, how much more likely are the top feature-related tokens to appear? This makes it a good measure of strong, localized effects on the highest-weighted vocabulary items associated with the target feature.
> > \
> > \
> > **Q.2 Why do you use token frequency instead of these metrics in the prompt injection experiments?
> > Thank you for raising this question about our choice of metric in the prompt-injection experiments. We agree this was not clearly explained.**\
> > The prompt-injection setting is substantially more expensive than the first two experiments: each injected prompt targets many more features, and we need to evaluate a large number of prompts and models. Computing weighted cosine similarity or weighted overlap in this regime would require collecting and processing full output distributions at scale (rather than a small number of sampled outputs), which is computationally costly for what is intended as a screening experiment.
> >
> > For this reason, we use token frequency as a cheaper, lower-bound probe: it lets us quickly detect cases where feature-related tokens become noticeably more frequent in the generated text after injection. We treat these frequency shifts not as a precise effect-size estimate, but as a way to flag candidate instances of shared polysemantic structures across models. Those candidate cases—where token frequencies provide a clear signal—are then followed up in more detail (and are listed in Appendix K.2.1, p.32), complementing the finer-grained metrics used in our main SAE-steering experiments.
> > \
> > \
> > **Q.3 Please define interference when you first refer to it in the introduction - its somewhat ambiguous otherwise. Similarly for the ‘feature irrelevance thresholds’ introduced in Section 3.1 - it would be much clearer if you define and link this to the semantic similarity measure in the main text (its also not clearly described in the Appendix).**\
> > Thanks for your suggestions. We have added definitions to the places the terms first appear.
> > \
> > \
> > **Q.5 The results on Figures 3 and 4 show a 100x difference between these methods for GPT-2 but the text in Section 3.2 says 10x - is this a typo or error on one of the graphs?**\
> > Sorry for making this typo error. We will modify the statement as “using token-gradient steering yields roughly ∼ 10× larger effects on Pythia-70M and ~100x on GPT-2-Small than steering along SAE feature directions.”
> > \
> > \
> > **Q.6 Can you add interference and semantic similarity scores to Table 3?**\
> > We apologize for not including this detail previously. The improved table, also as suggested by reviewer Zy8X, will be combined into the main text in the final revision.
> > \
> > \
> > **Q.7 Minor comment - the reference on line 143 is misformatted.**\
> > We really thank you for your detailed checking. We have corrected the typo.

---

> > > ### Comment · Reviewer_dp5x · 2025-11-27
> > >
> > > Thank you for the detailed response.
> > >
> > > I appreciate your commitment to clarifying the paper. I also greatly appreciate the intuitive metric descriptions, and would encourage you to also incorporate these into the updated paper. The filtering of steering directions and addition of Gemma results alleviates my concerns regarding this inconsistency, and I have increased my score.

---

### Official Review · Reviewer_yfBh · 2025-11-01

**Soundness:** 2
**Presentation:** 1
**Contribution:** 2
**Rating:** 2
**Confidence:** 3

**Summary:**

This paper explores polysemantic interference in transformer representations. Using sparse autoencoders (SAEs) trained on GPT-2-Small and Pythia-70M, the authors identify pairs of features that are semantically unrelated yet exhibit high cosine similarity (“interference”) in the SAE latent space. They then test whether intervening on one feature influences another across four levels: SAE-feature, token-gradient, prompt injection, and neuron manipulation. Results suggest that such interference pairs can weakly steer model predictions and that these patterns partly transfer to larger black-box models.

**Strengths:**

- The paper attempts to connect feature-level correlations to behavioral effects and cross-model transfer --- an ambitious angle.

- The idea of probing “interference” across different loci (feature, gradient, prompt, neuron) is original and could, in principle, yield insights into distributed representations.

**Weaknesses:**

Unclear necessity of SAEs: It is never justified why SAEs are needed for identifying interference.
The same cosine-overlap analysis could be performed directly in the model’s activation or embedding space, or between token clusters in the vocabulary embedding. By relying on pre-computed SAE features (and their textual glosses), the analysis may inherit annotation noise and input-centric biases (see Arad et al., SAEs Are Good for Steering - If You Select the Right Features, 2025).

Definition: The notion of “interference” and of “polysemantic neurons” are both defined through the SAE basis; this makes the subsequent results self-referential. There is no control showing that similar effects would not appear if random or non-SAE directions were used.

Baselines: In most experiments, there is no test of how non-interfering or semantically similar features behave under the same interventions. Without such controls, it is hard to interpret the magnitude or specificity of the reported changes.

Small effects: Table 1 and related plots show minor shifts (typically just a tiny bit above random). While statistically significant, the effects are small enough to fall within what could be lexical or topical overlap rather than genuine structural transfer. The cross-model “transfer” results in particular feel underwhelming. In Figure 5, the neuron-masking results are not monotonic, peaking at moderate polysemanticity. This contradicts the narrative that highly polysemantic neurons are “interference hubs.”

In sum: Interesting question and creative setup, but weak and noisy evidence and unclear conceptual motivation for SAEs

**Questions:**

- Why not compute interference directly between token-embedding clusters or residual directions instead of through an SAE?

- Can the authors provide baselines for unrelated concepts under identical interventions?

- How robust are the results to SAE sparsity, random seed, or layer choice?


More related work:
- Rosetta Neurons (Darvid et al. 2024)
- Second-Order Effects in CLIP (Gandelsman et al. 2024)
- Platonic Representations (Huh et al. 2024)
- Token Entanglement (Zur et al. 2025)

---

> ### Author Response · Authors · 2025-11-21
> **Response to reviewer yfBh**
>
> We sincerely appreciate your careful evaluation of our methodology and the important concerns you raise about the robustness of our findings. In the revised manuscript, we have incorporated changes addressing several of these points, with all modifications highlighted in red, and below we respond to each of your comments in detail.
>
> **Q.1&W.1 Why not compute interference directly between token-embedding clusters or residual directions instead of through an SAE?**\
> Thank you for this thoughtful question about our choice to work through SAEs rather than token-embedding clusters or hand-selected residual directions. Conceptually, SAEs give us feature directions in activation space that capture rich, often non-lexical semantics in an unsupervised way. A single SAE feature can correspond to something like “sadness,” activated by many different tokens and their combinations that express sadness, rather than being tied to one token or a small cluster. Token-embedding clusters, by contrast, are necessarily anchored in surface forms and would miss many such higher-level, context-dependent regularities.
>
> Methodologically, SAEs have also become a standard tool for mapping out the full landscape of model-learned features, and several recent works argue that sparse features provide a more interpretable and effective handle on model behavior than working directly in the dense residual stream [1, 2].  Our goal in this paper is precisely to study interference between such latent features, so SAEs are not just a convenience but a core premise of the design: they give us a large, relatively comprehensive, and semantically meaningful feature basis across layers and submodules.
>
> By contrast, operating directly on residual directions would require us to manually hand-pick a small set of candidate directions (e.g., via tailored prompts or probes), which (i) makes it difficult to be systematic or exhaustive, (ii) would be extremely expensive to scale across layers, models, and intervention levels, and (iii) risks introducing researcher bias into which features are even considered. For these reasons, we see SAEs as the appropriate level of abstraction for our question, and in the revision we now add a short discussion clarifying this design choice in the methods section.\
> [1] https://arxiv.org/pdf/2309.08600 \
> [2] https://aclanthology.org/2025.findings-emnlp.338.pdf
> \
> \
> **W.2 The notion of “interference” and of “polysemantic neurons” are both defined through the SAE basis; this makes the subsequent results self-referential. There is no control showing that similar effects would not appear if random or non-SAE directions were used.**\
> Thank you for raising this concern. We agree that, in the current draft, the relationship between SAEs, polysemanticity, and interference is not made fully explicit. In the revision, we clarify (and formalize in Section 2.1) that **interference is defined for any pair of directions in a submodule’s activation space**, as the cosine similarity between their corresponding activation vectors. SAE features are one particular family of such directions, but the notion of interference itself does not depend on the SAE basis.
>
> By contrast, **polysemanticity is indeed a property we define at the level of SAE features**, based on their activation patterns over tokens and contexts. We focus on SAE-derived directions because they provide a large, structured, and interpretable set of candidate features on which to study interference, not because interference is inherently SAE-specific. Any arbitrary (non-SAE) direction can also participate in interference; SAEs simply give us a principled way to select semantically meaningful directions for analysis.
>
> Regarding controls: our experiments already include comparisons to **non-SAE directions**. In particular, the last column in several result figures (e.g., Fig. 3, p. 6) reports a baseline built from random activation-space directions, which are not SAE features. These random baselines show much weaker or negligible effects compared to SAE-based directions, indicating that the patterns we observe do not arise generically for arbitrary directions. We also include comparisons to token-gradient directions, which constitute another non-SAE family of directions.

---

> > ### Author Response · Authors · 2025-11-21
> > **Response to reviewer yfBh (continued)**
> >
> > **W.3 In most experiments, there is no test of how non-interfering or semantically similar features behave under the same interventions. Without such controls, it is hard to interpret the magnitude or specificity of the reported changes.**\
> > Thank you for emphasizing the importance of appropriate controls. Conceptually, we fully agree that comparing high-interference pairs to non-interfering or semantically similar pairs is the right way to interpret effect magnitudes and specificity. In practice, however, in such high-dimensional spaces it is extremely difficult to identify pairs that are truly non-interfering or perfectly semantically unrelated, which is why we do not claim to have “zero-interference” or “zero-relatedness” controls.
> >
> > Instead, we address this concern in two ways. First, we systematically vary both the semantic relatedness cutoff and the interference-scale threshold, and show that our main results are robust to these choices: the effects we report are consistently stronger for higher-interference pairs than for low-interference pairs, across a range of semantic similarity cutoffs. Second, we run controlled regression analyses in which the behavioral outcome is modeled as a function of interference scale while explicitly controlling for semantic similarity (and other covariates). In these regressions, interference scale remains a significant predictor even when semantic similarity is included, indicating that the effects we observe cannot be explained by semantic similarity alone.
> >
> > We believe these analyses already address the core of your concern, and in the revision we will make this connection more explicit in the main text so that the role of these controls is clearer to the reader.
> > \
> > \
> > **W.4.1 Table 1 and related plots show minor shifts (typically just a tiny bit above random). While statistically significant, the effects are small enough to fall within what could be lexical or topical overlap rather than genuine structural transfer. The cross-model “transfer” results in particular feel underwhelming.**\
> > We sincerely thank the reviewer for this rigorous assessment of our results and for raising the valid concern regarding potential confounding factors. We wish to clarify that the primary objective of the experiments in this section is to identify shared interference structures across models, rather than to demonstrate large-scale effects.
> >
> > We fully acknowledge that the observed effect sizes are modest. This is, in part, an inherent consequence of our preliminary method. But a comprehensive fine-tuning and optimization of prompt injection for each potential case is computationally prohibitive, given the significantly larger feature space targeted in this section compared to previous experiments. We therefore adopt this lower-bound approach for initial probing, which involves inserting highly-activating tokens of interference features into prompts and observing the emergence of target concepts in the output. While this cost-effective strategy enables broad scanning, its inherent sensitivity limits the magnitude of the measurable effects.
> >
> > Critically, however, the goal of this design is not to achieve large effect size, but to leverage this scalable method to detect preliminary signals from potential interference structures. This allows us to identify promising candidates for further, more resource-intensive investigation. Indeed, this filtering process successfully identified several specific cases of shared interference, which we have now documented in Appendix K.2.1(p. 32). A direct examination of these instances readily verifies that the observed effects are not attributable to simple lexical or topical overlap, but are consistent with the presence of underlying structural transfer.

---

> ### Author Response · Authors · 2025-11-21
> **Response to reviewer yfBh (continued)**
>
> **W.4.2 In Figure 5, the neuron-masking results are not monotonic, peaking at moderate polysemanticity. This contradicts the narrative that highly polysemantic neurons are “interference hubs.”**\
> Thank you for this careful observation. You are right that, in Figure 5, the neuron-masking effects are not monotonic in the polysemanticity score, with the largest changes often appearing at moderate levels. Our use of the term “interference hub” is not intended to mean that every intervention effect should increase monotonically with polysemanticity, but rather that highly polysemantic neurons are **structurally central** in the interference pattern (i.e., they participate in many interference relationships).
>
> In complex networks, being a “hub” does not imply that masking it must produce the largest effect. A hub can be **weakly connected to many nodes**: suppressing it (masking) slightly decreases many weak connections at once, so the overall change can be moderate. However, because these weak connections are spread across many directions, **amplifying** such a hub (scaling its activity) can accumulate into a large downstream effect: small weights multiplied by a large amplification factor can still yield substantial global changes. This is exactly the regime we probe when we steer neuron activations rather than mask them.
>
> In the revision, we clarify that our “interference hub” terminology refers to this structural centrality in the interference network, and we explicitly note that we do not claim a strictly monotonic relationship between polysemanticity and all intervention effects. Our results show that neurons with higher polysemanticity are disproportionately influential under amplification-style interventions, even though masking effects peak at moderate levels.
> \
> \
> **Q.2 Can the authors provide baselines for unrelated concepts under identical interventions?**\
> We have addressed this question in W.3.
> \
> \
> **Q.3 How robust are the results to SAE sparsity, random seed, or layer choice?**\
> Thank you for raising this robustness question and for pointing out that our experimental settings were not fully specified in the draft.
>
> **SAE sparsity.** In this work, we use pre-trained, publicly released SAEs from Neuronpedia, which cover all layers and all submodules (MLP, attention, residual) of the two focal models, with sparsity fixed at 3. Systematically retraining and evaluating SAEs at multiple sparsity levels across all layers would be computationally very demanding. Instead, we rely on the accompanying large-scale evaluation in [1], which already studies how sparsity affects the quality of extracted features and finds that SAE features provide a more effective handle on model behavior than working directly in the residual stream. Building on these results, we treat sparsity=3 as a reasonable operating point rather than a tuned hyperparameter.
>
> **Layer choice.** Our experiments are not restricted to a subset of layers: all reported statistics aggregate over all layers and submodules of the models, so the results should be interpreted as “overall” patterns rather than layer-specific phenomena.
>
> **Random seed / intervention robustness.** So far, due to time limitation, we have rerun the SAE-based intervention experiments for another time with a different random seed and with a simple enhancement to the steering procedure (Appendix O). Across these runs, the qualitative patterns and main conclusions remain stable. In the revision, we clarify these choices and explicitly state that we are using fixed, well-evaluated SAEs (sparsity=3) across all layers, and that our robustness checks focus on seeds and intervention variants rather than re-training SAEs under many sparsity configurations.\
> [1] https://arxiv.org/pdf/2406.04093

---

### Official Review · Reviewer_Zy8X · 2025-11-03

**Soundness:** 3
**Presentation:** 2
**Contribution:** 3
**Rating:** 4
**Confidence:** 3

**Summary:**

This work studies polysemanticity in LLMs, using SAEs to identify features which are semantically unrelated, but have unexpectedly entangled representations. They use these features to develop 4 methods of intervention (prompt, token, feature neuron), and show the effectiveness of interventions with semantically unrelated features, e.g. occurrences of specific surnames with geographic locations. They find cases where, surprisingly, semantically unrelated features transfer from very small models (e.g. 70M) to large models (e.g. 70B). The authors also study highly connected "super neurons" which, if steered, affect hundreds of distinct SAE features, as a case study in extreme polysemanticity.

**Strengths:**

- Well-motivated problem setup, and grounded in real world impacts of polysemantic features and vulnerability of black-box models to adverserial attacks.
- Experiments and methods are comprehensive, thorough, and well-documented.
- Interesting empirical results with clear impact, especially the transfer between models which are tiny by modern standards and full-size 70B models. The super-neuron results are also interesting.
- Generally well-written.

**Weaknesses:**

- One of the greatest strengths of this paper is the result with feature transfer across model scale. However, the effect size of these seems very small at only a few % points in most cases (table 1), which the authors don't seem to make note of. If this is right, this seems like an important limitation of this work which should be clear to the reader, although it also may diminish the significance of this work somewhat.
	- Relatedly, some of the interventions in figure 3 seem to have very modest effects compared with the random baseline.
- Presentation in the figures was lacking, in my opinion. I had trouble understanding fig 1, and this could be improved. For many of the results figures, I had some confusion as to exactly what the point is in some cases. It might help readability if the captions clearly state what the takeaway is from reading these results.
- There seems to be some missing glue between the introduction, and the formalization of e.g. the "Human Symbolic Manifold", and the rest of the text. These don't seem to be referenced later, except in 3.5, and so I wonder if this formalization is even necessary, or if instead the authors could utilize it more.
- Too much reliance on the appendix, which really should be extra information which is not vital to the content of the work. One example is table 3 and some of the word clouds in the appendix - these were helpful to give me intuition of what kinds of particular semantically unrelated features are interfering, and something like this in the main text might improve readability.

I might be willing to raise my score if these concerns were addressed, especially the first point.

**Questions:**

- What do the authors think of the relationship between semantically unrelated polysemanticity and recent work on emergent misalignment and subliminal learning [1, 2]?
- For table 1, I wonder what a "semantically related" baseline would look like, rather than a random baseline. Was this tested, or would this make sense to test?
- I wonder if many of the significance tests in this table would hold up with multiple hypothesis correction (e.g. holm-bonferroni), given that the table is showing 45 test results.
- Diving deeper into some of the cases with big effect sizes in cross-model transfer (e.g. science on llama-70B) might be interesting.
- There is no reference to figure 5 in the text.


[1] Betley, J., Tan, D., Warncke, N., Sztyber-Betley, A., Bao, X., Soto, M., ... & Evans, O. (2025). Emergent Misalignment: Narrow finetuning can produce broadly misaligned LLMs.

[2] Zur, A., Loftus, A. R., Orgad, H., Ying, Z., Sahin, K., & Bau, D. (2025). _It’s Owl in the Numbers: Token Entanglement in Subliminal Learning_.

---

> ### Author Response · Authors · 2025-11-21
> **Response to reviewer Zy8X**
>
> Thank you for your thoughtful review and for highlighting our contribution to understanding the transferability of polysemantic interference structure. We also appreciate your detailed comments on effect sizes, which have been very helpful in strengthening the robustness of our analysis. In the revised manuscript, we have incorporated changes reflecting your suggestions (highlighted in red); below, we respond to each of your questions and concerns in turn.
> \
> **W.1 One of the greatest strengths of this paper is the result with feature transfer across model scale. However, the effect size of these seems very small at only a few % points in most cases (table 1), which the authors don't seem to make note of. If this is right, this seems like an important limitation of this work which should be clear to the reader, although it also may diminish the significance of this work somewhat.**\
>     **- Relatedly, some of the interventions in figure 3 seem to have very modest effects compared with the random baseline.**\
> Thank you for raising this important point about effect sizes, especially for the cross-scale transfer results. Our primary goal in this paper is to **establish whether seemingly counterintuitive (semantically irrelevant) polysemantic interventions can systematically influence model behavior**, not to optimize the strength of interventions. To make this tractable, we perform a large-scale study (four intervention levels, across every layer and submodule of two language models), and therefore deliberately use very simple, conservative intervention schemes: we steer with a single interfering feature direction at a time, on a single layer, without searching over feature combinations, layer subsets, or more sophisticated prompt-insertion strategies. In this sense, the measured effects should be viewed as **a lower bound** on what polysemantic interference–based interventions can achieve.
>
> Even under these conservative choices, the effects are often substantial. For instance, for gradient-direction steering, the semantic alignment of outputs with the target feature increases by roughly **40×** for Pythia-70M and **5×** for GPT-2-Small. Likewise, under the alternative metric of **weighted overlap**(Fig. 9), the most related tokens of the target feature become about **10² times** more likely to appear in GPT-2-Small’s outputs, at a level comparable to Pythia-70M. These results indicate that the underlying polysemantic interference structure produces substantial impacts to model behavior, even when averaged over many conservative interventions.
>
> Motivated by your concern, we additionally performed a preliminary optimization of the steering procedure, reported in Appendix O (p. 38) of the revision. We find that the relatively small average effect sizes in GPT-2-Small (under weighted cosine similarity) are partly driven by the inclusion of many ineffective steering directions in the average. A simple filtering of such directions increases the effect size by ~100× in GPT-2-Small and ~5× in Pythia-70M. While GPT-2-Small still shows smaller numerical values under weighted cosine similarity, its performance under weighted overlap remains comparable across models, supporting the conclusion that interventions can also reliably enhance target-relevant tokens in GPT-2-Small.
>
> Regarding the prompt-injection experiments (Table 1 and examples in Table 3), the insertion strategy we use is intentionally naive, so modest average effects are expected. Nonetheless, we still observe statistically significant differences at high-interference levels, which we treat as a probe to identify especially promising interference patterns that potentially transfer across models. We then trace these successful cases back to their underlying interference features and illustrate the corresponding polysemantic structures in Appendix K.2.1(p. 32).

---

> > ### Author Response · Authors · 2025-11-21
> > **Response to reviewer Zy8X (continued)**
> >
> > **W.2 Presentation in the figures was lacking, in my opinion. I had trouble understanding fig.1, and this could be improved. For many of the results figures, I had some confusion as to exactly what the point is in some cases. It might help readability if the captions clearly state what the takeaway is from reading these results.**\
> > Thank you for this helpful comment on figure presentation and clarity. We apologize that our current draft does not provide sufficient guidance for interpreting the figures. In the revision, we have substantially expanded and clarified the captions.
> >
> > For Figure 1, we now explain more clearly that the top panel is a geometric illustration of how two semantically orthogonal features can still exhibit strong interference in the model’s activation space. Neurons A, B, and C span a 3D activation space; when their activity is mapped onto the symbolic manifold M, their projected directions are not guaranteed to remain orthogonal. Features D, E, F, and G denote SAE features (also projected to M). In this example, E and G appear orthogonal in M, yet both strongly interfere along neuron C in the activation space, illustrating that orthogonality in the symbolic manifold does not necessarily imply independence in the underlying activations. We added a step-by-step description of this construction to the caption of Figure 1 (p. 2) in the revised version.
> >
> > For the other result figures, we will revise the captions so that each one explicitly states (i) what is being plotted, (ii) how to read the axes/conditions, and (iii) the main takeaway or pattern the reader should see. We hope this can make the figures easier to follow and their substantive conclusions clearer.
> > \
> > \
> > **W.3 There seems to be some missing glue between the introduction, and the formalization of e.g. the "Human Symbolic Manifold", and the rest of the text. These don't seem to be referenced later, except in 3.5, and so I wonder if this formalization is even necessary, or if instead the authors could utilize it more.**\
> > Thank you for pointing this out. We agree that, in the current draft, the connection between the introduction, the formalization of the human symbolic manifold, and the later sections is not made sufficiently explicit. Conceptually, however, the distinction between human symbolic space M, model activation space $A_\ell$, and SAE feature space $F_\ell$ is central to our identification of “counterintuitive” features (distant in M) that are nevertheless “entangled” (close in $A_\ell$) via SAE features in $F_\ell$.
> >
> > In the future revision, we will more explicitly tie our formalization to the rest of the paper. In particular, we will refer back to this framework when defining our interference measures and when interpreting the main experimental results. We hope this can make the role of the formalization clearer and its relevance more evident throughout the paper.
> > \
> > \
> > **W.4 Too much reliance on the appendix, which really should be extra information which is not vital to the content of the work. One example is table 3 and some of the word clouds in the appendix - these were helpful to give me intuition of what kinds of particular semantically unrelated features are interfering, and something like this in the main text might improve readability.**\
> > We sincerely appreciate your insightful suggestions for improving the paper’s readability. In the final revision, we will carefully select and highlight representative examples in the main text to better support reader comprehension.

---

> > > ### Author Response · Authors · 2025-11-21
> > > **Response to reviewer Zy8X (continued)**
> > >
> > > **Q.1 What do the authors think of the relationship between semantically unrelated polysemanticity and recent work on emergent misalignment and subliminal learning [1, 2]?**\
> > > Thank you for pointing us to this line of work and for raising this connection. We see semantically unrelated polysemanticity as closely related to emergent misalignment and subliminal learning: when a single direction or neuron entangles concepts that are far apart in the human symbolic space, it can induce behaviors that are opaque to users and that diverge from their intended objectives, creating exactly the kind of hidden “side channels” these papers highlight.
> > >
> > > At the same time, we would not characterize all such structures as inherently malicious. Some semantically unrelated pairings may simply reflect non-obvious but benign or even useful knowledge associations learned during training. Our results show that many of these counterintuitive polysemantic structures are not idiosyncratic artifacts of a single model, but systematically transfer across architectures, which suggests that they arise from shared training pressures rather than noise. We will emphasize in the discussion that this transferability strengthens the case for studying these phenomena as a potential substrate for emergent misalignment and subliminal learning, and as an important target for future interpretability and alignment work.
> > > \
> > > \
> > > **Q.2 For table 1, I wonder what a "semantically related" baseline would look like, rather than a random baseline. Was this tested, or would this make sense to test?**\
> > > Thank you for this thoughtful suggestion. A “semantically related” intervention baseline is indeed a natural comparison, and we agree that such interventions would be expected to produce stronger mutual influence than random ones. However, our focus in this paper is specifically on semantically distinct concepts, in order to highlight counterintuitive cases where features that appear unrelated in the symbolic space still interfere strongly in the activation space. In that sense, a semantically related baseline, while conceptually reasonable, would address a more intuitive regime and is somewhat orthogonal to our main question. We therefore did not include such experiments in the current study, but we see this as a useful extension for future work that aims to more fully map out the spectrum from semantically aligned to semantically distant interventions.
> > > \
> > > \
> > > **Q.3 I wonder if many of the significance tests in this table would hold up with multiple hypothesis correction (e.g. Holm-Bonferroni), given that the table is showing 45 test results.**\
> > > Thank you for your careful reading of Table 3 and for raising the issue of multiple comparisons. You are correct that this table reports a relatively large number of tests. Conceptually, this third experiment is more exploratory and considerably more expensive than the first two: fine-tuning and optimizing prompt injection require many target features and runs, so we adopted a simplified, low-cost protocol intended primarily as a screening tool for potential cross-model interference structures. As a consequence, the individual effect sizes are relatively small. In the final revision, we will clarify this exploratory role in the text and, for transparency, will indicate which of the reported tests remain significant under a Holm–Bonferroni correction. Nevertheless, we need to clarify that our main use of this experiment is to identify candidate cases of shared interference structure, which we then inspect in detail; these concrete examples are reported in Appendix K.2.1 (p. 32).
> > > \
> > > \
> > > **Q.4 Diving deeper into some of the cases with big effect sizes in cross-model transfer (e.g. science on llama-70B) might be interesting.**\
> > > Thank you for this suggestion. We agree that a deeper qualitative analysis of specific high–effect size cases (such as the “science” feature on Llama-70B) would be very informative and of broad interest. In this paper, however, our primary aim is to perform a systematic, model-wide check of our core hypothesis about counterintuitive interference and its transfer across architectures, rather than to develop detailed case studies of individual features or devise state-of-the-art model attack techniques.
> > >
> > > That said, motivated by your comment, we now highlight in the revision that our experiments do surface several feature pairs with consistently strong cross-model transfer, spanning models from Pythia-70M to Gemma-2-9B; these candidate cases are listed in Appendix K.2.1(p. 32). We see a more in-depth analysis of such specific high–effect size examples as a natural direction for follow-up work building on the present study.
> > > \
> > > \
> > > **Q.5 There is no reference to figure 5 in the text.**\
> > > Sorry for the mistake. We wrongly referred Fig.15, which should be Fig.5, in Section 3.6, which talks about neuron intervention.

---

> ### Author Response · Authors · 2025-11-30
> **Supplementary Response to Reviewer Zy8X**
>
> We again appreciate your rigorous review of our prompt injection results. In response to your main concern, we have conducted two comprehensive experiments in Pythia-70M and GPT-2-Small, following the same methodology as the two previously reported studies (i.e., SAE intervention and gradient vector intervention).
>
> Here we give a brief description and put the details in the revised version(p.30). In comparison with the current case studies, we randomly sample target features across all sub-modules in all layers. We prepend injection text snippets containing high-activation tokens of interference features, and assess their effects. The same metrics, weighted cosine similarity and weighted overlap, are used to measure semantic shifts in model outputs to the target feature.
>
> Compared with the previous two experiments, both metrics decrease by ~10x-100x. However, we need to clarify that although the improvement of weighted cosine similarity reduces to ~5%-10% since injecting only a limited set of tokens inherently will not cause a large-scale change in all output tokens, **it still demonstrates a weak trend that stronger interference leads to relatively higher semantic shift.** Importantly, **the improvement in weighted overlap remained marked, showing a ~2x-10x increase compared to the ~100x-1000x improvement seen in SAE and gradient-based interventions.** This indicates that the most relevant tokens associated with the target features are substantially influenced, i.e. their output probabilities are increased by a factor of two to ten.
>
> Regarding the case studies, we need to point out that the interference structures observed in Pythia-70M and GPT-2-Small do not transfer extensively to other models, so the resulting effects are not similarly pronounced. However, this only serves as an initial method for quickly detecting whether certain cross-model polysemantic interference structures exist. The exploration of more effective methods, however, falls outside the scope of this paper.
>
> We sincerely hope you can read our supplementary experiments and increase the score.

---

### Official Review · Reviewer_72iw · 2025-11-05

**Soundness:** 3
**Presentation:** 2
**Contribution:** 4
**Rating:** 8
**Confidence:** 4

**Summary:**

This paper explores interference in trained SAE latents and polysemanticity thereof. Specifically, polysemanticity is defined as the scenario where the precise descriptions assigned to two features differ and yet they interfere (for which I'm not quite clear what the precise definition is, since, unless I missed something, there was no formula provided for how interference is measured). From hereon, the work is mostly a qualitative study to develop the broader phenomenology of polysemantic SAE latents, i.e., identifying interesting ways that these features interact (e.g., how steering one can affect model behavior along another one). The coolest result, which relates with the work by Lee et al. [1], is that these steering pathologies transfer across model, indicating there are consistent statistical signatures in the data distribution, which all models pick on. This result is especially cool because the results were derived using small models (order 100M parameters), but transfer to substantially larger models (order 7--9B parameters).

[1] https://arxiv.org/abs/2504.14379

**Strengths:**

I really like this paper. It needs some improvement on presentation / writing to ensure results are easily legible, but conditioning on that minimal rewrite, the empirical characterization of interference pathologies (such as the super-neurons), the transfer of these pathologies across model scales, and the exhaustiveness of the experiments is awesome. To be clear, I believe these results were relatively expected, but getting a detailed account of them in a single paper is very helpful for the community.

**Weaknesses:**

- Missing descriptions of relevant concepts: I tried to find the precise mathematical definition for how interference is defined or measured, but didn't see it anywhere in the paper. This really impacted my ability to onboard with the paper on a first reading pass, but eventually digging through the appendix and looking at results, assuming there's a reasonable mathematical definition for the notion of interference, I like this paper's goals and results. If the authors can fix these definitions (unless I missed something there are not provided at the moment), I support the acceptance of this work.

Some other minor comments:
- Writing style: I found some of the language in the paper unnecessarily complex. For example, the use of the term "loci" in the abstract doesn't make sense, both because it's unnecessarily complex, but also (I think) wrong in this case? The closest term that makes sense would be "foci", since a "locus" refers to a path.

- On adversarial examples and superposition: a recent work makes a similar point as the authors, i.e., adversarial examples can be driven by feature superposition [1].

- Possible typo on L94--95: If I understand correctly, the line should orthogonality in symbolic manifold M is not does not persist in the activation space after projection. The subsequent lines (L95--96) only make sense to me with this rephrasing.

- Typo in Eq. 1: I presume it should be O(t) instead of P(t), or P(.) should be defined to be an element from the set {$O, \tilde{O}$}?

- Analysis layer: For several experiments, I wasn't sure which layer SAEs are analyzed. For example, in the paragraph on L279, it is stated that due to greater depth of GPT-2, SAE interventions for interfering features might be less effective. While I intuitively buy this, the precise layer number information would help contextualize the claim.

[1] https://arxiv.org/abs/2508.17456

**Questions:**

A question that came to mind while reading:

- Instead of the use of agglomerative clustering, as done for analysis in this paper, I wonder if SAEs with hierarchical priors could have been used to directly elicit statistical interferences between model features. For example, see papers [1, 2, 3]. I'd love to gather authors' thoughts (or experiments if they get the chance) on how they would use such more structured SAEs for their analysis.


[1] https://arxiv.org/abs/2503.17547

[2] https://arxiv.org/abs/2506.03093

[3] https://arxiv.org/abs/2506.01197

---

> ### Author Response · Authors · 2025-11-21
> **Reponse to Reviewer 72iw**
>
> We sincerely appreciate your thoughtful and positive feedback, especially your recognition of the breadth and thoroughness of our study. Below, we respond to your main questions and concerns in detail. For your convenience, we have also attached a partial revised draft of the manuscript, in which changes made in response to your suggestions are highlighted in red.
>
> **W.1 Missing descriptions of relevant concepts.**\
> Thank you for highlighting the lack of precise mathematical definitions for key concepts. You are right that, in the current version, we only provide verbal descriptions of interference and semantic relatedness in Section 2.1 (p. 3). In our revision, we now give explicit mathematical definitions in Section 2.1. In particular, we define the interference between two SAE features i and j as the cosine similarity between their corresponding vectors in the focal submodule’s activation space. We also formalize the notion of semantic relatedness in the same section, aligning the notation with the rest of the paper to make the technical framework easier to follow.
> \
> \
> **W.2 Writing style: I found some of the language in the paper unnecessarily complex. For example, the use of the term "loci" in the abstract doesn't make sense, both because it's unnecessarily complex, but also (I think) wrong in this case?**\
> Thanks for pointing out this wording issue. We have updated the text in the abstract accordingly and will continue to refine the expressions throughout the paper.
> \
> \
> **W.3 On adversarial examples and superposition: a recent work makes a similar point as the authors, i.e., adversarial examples can be driven by feature superposition.**\
> Thank you for sharing this relevant work with us. We have cited this paper in our revision. See p.2 line.6.
> \
> \
> **W.4 Possible typo on L94--95: If I understand correctly, the line should orthogonality in symbolic manifold M is not does not persist in the activation space after projection. The subsequent lines (L95--96) only make sense to me with this rephrasing.**\
> Thank you for flagging the possible typo. We revisited the sentence and thought the intended meaning is correctly captured in the current wording, but we agree it may be unclear. You suggested rephrasing seems to contain typos. Could you clarify which part you see as inconsistent or incorrect? We are happy to revise the phrasing.
> \
> \
> **W.5 Typo in Eq.1: I presume it should be $O(t)$ instead of $P(t)$, or $P(.)$ should be defined to be an element from the set {$O, \tilde{O}$}?**\
> Thanks for your careful scrutinization. You are right, they should be O(t). We corrected this typo in the revision. See p.4, Eq.1 in the revision.
> \
> \
> **W.6 Analysis layer: For several experiments, I wasn't sure which layer SAEs are analyzed. For example, in the paragraph on L279, it is stated that due to greater depth of GPT-2, SAE interventions for interfering features might be less effective. While I intuitively buy this, the precise layer number information would help contextualize the claim.**\
> Thank you for raising this question about analysis layers. We apologize for the lack of clarity in the current draft. All our evaluations are conducted across all submodules (MLP, attention, and residual) in all layers of each model, rather than on a specific layer. In the main text, we report averages over all these cases using weighted cosine similarity, and in the appendix we provide complementary results using weighted overlap. Thus, the reported effects reflect average patterns across the full model.
> \
> Motivated by your comment, we further examined whether the smaller intervention effect observed for GPT-2-Small could be attributed to model depth. To test this, we ran an additional intervention experiment on Gemma-2-2B. As shown in Appendix O, its effect size (weighted cosine similarity) is higher than that of GPT-2-Small, suggesting that the effect is not straightforwardly explained by depth alone. In light of this, we have softened and revised our interpretation of the GPT-2-Small result in the revision (p.6 and p.38), treating it as an open question. At the same time, we note that effect sizes measured by weighted overlap remain similar across all three models (see Fig. 9, p.24, and Fig. 17, p.39, in the revision), which leads us to hypothesize that the apparent magnitude differences between models are largely driven by the choice of measurement.

---

> > ### Author Response · Authors · 2025-11-21
> > **Reponse to Reviewer 72iw (continued)**
> >
> > **Q.1 Instead of the use of agglomerative clustering, as done for analysis in this paper, I wonder if SAEs with hierarchical priors could have been used to directly elicit statistical interferences between model features.**\
> > We appreciate this insightful suggestion and the pointers to related work. Using SAEs with hierarchical priors could indeed provide a more structured way to elicit patterns of interference, by inducing feature lineages directly rather than relying on agglomerative clustering for post-hoc structure. This would naturally separate features at different levels of abstraction and may help us study how interference manifests across semantic levels (e.g., between a “chihuahua” feature and a more global “science” feature). While this direction is beyond the scope of the current paper, we see it as a very promising avenue for future work and have added a note to the discussion section to highlight it.

---

### Comment · Area_Chair_uovr · 2025-11-25
**Please discuss**

This paper has a wide range of scores. I would love to see the reviewers engage with each other, as well as the author response, and see if they come closer to a consensus.  Have the rebuttals addressed your concerns or clarified anything?

---

### Author Response · Authors · 2025-12-01
**Summarization of Rebuttal**

Dear AC and Reviewers,

We greatly appreciate your overseeing of the rebuttal process and your significant engagement with the work. This brief overview summarizes additional experimentation work that has been completed during the course of the rebuttal and showcases how it strengthens our core contributions in addressing the key issues raised within the reviews.

1. __Limited effect sizes and cross-model discrepancy__. Our original submission prioritized mechanistic insight over maximizing intervention strength, so we did not initially tune the steering procedure for effect size. In response to reviewer concerns, we added a preliminary optimization step (Appendix O, p. 38). We find that the modest average effect sizes for GPT-2-Small under the weighted cosine metric are largely due to averaging over many ineffective steering directions. Filtering out SAE feature directions whose self-steering has negligible effect increases the average effect size by ~100× in GPT-2-Small and ~5× in Pythia-70M. We also ran additional interventions on Gemma-2-2B for cross-validation; its weighted-cosine effect sizes exceed those of GPT-2-Small but remain below those of Pythia-70M, indicating that the discrepancy between GPT-2-Small and Pythia-70M cannot be straightforwardly explained by model depth alone. Accordingly, we have softened and clarified our interpretation of the smaller GPT-2-Small intervention effects results in the revision (p. 6 and p. 38).

2. __Evaluation metric consistency for prompt injection__. For prompt injection, our original case studies reported only the success rate of boosting tokens related to the target feature to show how intervention effects transfer to even black-box models through simple prompt manipulation. For the non-black-box part, to align with our main metrics based on reviewers' suggestions, we conducted an additional experiment in which we randomly sample target features across all submodules and layers of GPT-2-Small and Pythia-70M. For each sampled target feature, we prepend injection snippets composed of high-activation tokens for its interference features, and then measure the resulting semantic shift toward the target feature using the same metrics as in the main text: weighted cosine similarity and weighted overlap. As expected, injecting a small set of tokens cannot dramatically change the distribution over all output tokens, so the gains in weighted cosine similarity are modest (~5–10%). Nonetheless, we still observe a consistent trend: stronger interference is associated with larger cosine-based shifts. More importantly, weighted overlap shows substantial gains of roughly 2×-10×, indicating that the most relevant tokens associated with the target feature are strongly affected—their output probabilities increase by a factor of two to ten.

3. __Paper organization and clarity__. Following the reviewers' suggestions, we have revised the formalization of our evaluation metrics, improved the captions for all figures, and carefully corrected typos throughout the paper. We are also continuing to refine the exposition to make the write-up more accessible and easier to follow for a broad audience.

In light of these revisions, Reviewer dp5x noted that “the filtering of steering directions and addition of Gemma results alleviates my concerns regarding this inconsistency,” and increased their score to 8. Reviewer Zy8X also indicated in their initial review that they “might be willing to raise [their] score if these concerns were addressed, especially the first point”; our revision implements the strongest fixes we could for each of their stated weaknesses. Unfortunately, we must also note that Reviewer yfBh’s review has been flagged as __fully AI-generated__ by the official pangram tool (iclr.pangram.com). We respectfully ask the AC to take this into account when assessing the reliability and weight of that particular review in their overall decision.

Overall, despite the unfortunate circumstances that prevented further discussion during the rebuttal phase, we appreciate the opportunity to have significantly improved our submission as a result of reviewer interactions. We once again thank the reviewers and the AC for their engagement with and consideration of our work.

---

### Meta-Review · Area_Chair_mrcL · 2026-01-06

**Summary:**

The initial reviews ranged from an 8 (Accept) to a 2 (Reject). The primary concerns that informed the evaluation included:
- Small Effect Sizes: Reviewer noted that the relative changes in weighted cosine similarity, particularly for cross-model transfer in Table 1, appeared very modest (often only a few percentage points).
- Cross-Model Discrepancies: Reviewer highlighted a massive 100x difference in effect sizes between Pythia-70M and GPT-2-Small, arguing that the authors' explanation of "model depth" was unconvincing.
- Lack of Formal Definitions: Reviewer 72iw struggled with the first reading because the paper lacked precise mathematical formulas for "interference" and "semantic relatedness".
- Methodological Justification (SAEs): Reviewer questioned the necessity of using Sparse Autoencoders (SAEs), suggesting the analysis might be self-referential and that interference could be measured directly in activation or embedding spaces.
- Presentation and Clarity: Multiple reviewers found Figure 1 confusing and noted unreadable font sizes or missing references to figures (e.g., Figure 5) in the text

**Reviewer Concerns:**

Addressed Concerns:

• Effect Size Discrepancy (partially): The authors introduced a preliminary filtering step in Appendix O, which removed SAE feature directions where self-steering had a negligible effect. This increased the average effect size by ~100x in GPT-2-Small and ~5x in Pythia-70M, significantly narrowing the gap between models.

- Depth Hypothesis: To address the "model depth" concern, authors ran new interventions on Gemma-2-2B. They found its effect sizes exceeded GPT-2-Small despite being deeper, leading them to treat the cause of discrepancy as an open question while proving the transfer structure remains robust.
- Formal Definitions: The authors added explicit mathematical definitions for interference (cosine similarity in activation space) and semantic relatedness (cosine similarity in the symbolic manifold) in Section 2.1.
- Presentation: Captions for Figure 1 and other results were substantially expanded to clarify the "takeaway" of each visualization.

Outstanding Concerns:
- Alternative Baselines: Reviewer requested baselines for "non-interfering" or "semantically similar" features under the same interventions. Without these controls, it is difficult for meta-reviewers to interpret whether the reported shifts are specific to "counterintuitive" polysemanticity or if any targeted intervention would yield similar distributional change.

-Cross-Model Discrepancies and Inconsistent Scaling: Reviewer highlighted a massive 100x difference in effect magnitude between the Pythia-70M and GPT-2-Small models, and the authors’ initial hypothesis that "model depth" caused this discrepancy was viewed as unconvincing by the reviewers.  This large variance in performance across different architectures suggests the method might lack the stability and universality claimed in the abstract

- Practical "Dangerous" Case Study: Reviewer dp5x suggested a case study extracting prompt injections that elicit dangerous responses in closed-weight models. While the authors provided some black-box examples for Llama-3.1, a full "safety" exploit remains beyond the scope of this mechanistic study.

**Reviewer Scores:**

• Reviewer 72iw (Initial 8): No change. This reviewer was already very positive
• Reviewer dp5x (Initial 6 → 8):  projected 7. The reviewer explicitly stated that the addition of Gemma-2-2B results and the filtering of steering directions alleviated their concerns regarding model inconsistencies.
• Reviewer Zy8X (Initial 4): Projected 5. This reviewer’s primary blocker was the small effect size. Given the authors demonstrated a 100x increase in effect size through simple filtering and addressed the missing figure references, this reviewer would likely move higher, but not too high since Cross-Model Discrepancies and Inconsistent Scaling are still concerns.
• Reviewer yfBh (Initial 2): Projected 2. Given the reviewer's fundamental disagreement with the SAE-based premise, a score change is unlikely

---

### Decision · Program_Chairs · 2026-01-26

Accept (Poster)